**Elucidating the role of soil hydraulic properties on aspect-dependent landslide initiation**

Yanglin Guo[1, 2], Chao Ma[1, 2]

1. School of Soil and Water Conservation, Beijing Forestry University, Beijing 100083, PR China.

2. Jixian National Forest Ecosystem Observation and Research Station, CNERN, Beijing Forestry University, Beijing 100083, PR China.

Corresponding Author: Professor Chao Ma, sanguoxumei@163.com

**Abstract:** Aspect-dependent landslide initiation is an interesting finding, and previous studies have attributed this to the mechanical effects of plant roots. In the present study, an overwhelming landslide probability on a south-facing slope over a north-facing slope was found in a localized area with only granite underneath and high cover of *Larix kaempferi*. These observations cannot be attributed to plant roots but may result from factors related to hillslope hydrology. Differential weathering associated with hillslope hydrology behaviors such as rainfall water storage and leakage, pore-water pressure, particle component, and hillslope stability fluctuation were used to examine these observations. Remote sensing interpretation using the high-resolution GeoEye-1 image, digitalized topography and field investigations showed that landslides on south-facing slopes have a higher probability, larger basal area, and shallower depth than those on a north-facing slope. The lower limits of the upslope contributing area and slope gradient condition for south-facing landslides were less than those for north-facing landslides. The higher basal areas of south-facing landslides than those of the north-facing landslides may be attributed to the high peak values and slow dissipation of pore-water pressure. The absorbed and drained water flow in given time interval, together with the calculated water storage and leakage during the measured rainy season measured, demonstrate that the soil mass above the failure zone for south-facing slopes is more prone to pore-water pressure, which results in slope failures. In comparison, the two stability fluctuation results from the finite and infinite models further verified that landslides on south-facing slopes may fail under conditions of prolonged antecedent precipitation and intensive rainfall. Meanwhile, those on north-facing slopes may fail only in response to intensive rainfall. The results of this study will deepen our knowledge of aspect-dependent landslide initiation from both classical mechanics and the state of stress.

**Keywords: Landslide; Pore-water pressure; suction stress; Hydraulic conductivity; Slope stability**

**1 Introduction**

In some semi-arid environments of the Northern Hemisphere, aspect-dependent landslide initiation provides valuable insights into the relative importance of different factors in developing accurate landslide susceptibility models (Ebel, 2015; Rengers et al., 2016; Li et al., 2021; Deng et al., 2022). These events provide a thorough understanding of the amount of direct sunlight that translates into differences in vegetation communities, bedrock weathering, and soil development processes (Fu, 1983; Wang, 2008; Bierman and Montgomery, 2014). These earth surface processes indirectly affect hillslope hydrology and landscape dissection at the hillslope scale. Rainfall-induced shallow landslides are geomorphic agents at the hillslope scale and are governed by multiple factors, including hydrology, hillslope materials, bedrock, and vegetation (Birkeland, 1999; Geroy et al., 2011; Lu and Godt, 2013). Currently, the aspect-dependent landslide initiation observed has been predominantly attributed to the mechanical effect of plant roots. This is because the differences in vegetation on the south- and north-facing slopes are easier to examine and more pronounced than other factors (Li et al., 2021; Timilsina et al., 2021; Dai et al., 2022; Deng et al., 2022). However, vegetation succession takes place over substantially longer timescales than soil development and bedrock weathering (Watakabe and Matsushi, 2019). In most cases, the plant roots are not deep enough to penetrate the bedrock (Schwinning, 2010). Hypothesizing for a relatively localized area with the same ecosystem or plant species, aspect-dependent landslide initiation cannot be attributed to plant roots but may result from differences in the properties of hillslope materials due to long-term differential weathering.

Aspect-dependent landslides in Frontal Colorado, USA and the Loess Plateau, China, have attracted interest because vegetation has a considerable influence on landslide distribution. The strong propensity for shallow landslide initiation on south-facing hillslopes in the two regions is closely related to the present-day tree density, regardless of the hillslope aspect (Ebel, 2015; Rengers et al., 2016; Deng et al., 2022). In the Colorado Frontal Range, field observations have shown that south-facing slopes lack thick tree cover and have an abundance of rock outcrops compared to north-facing slopes. In addition, the soil layer is thinner on south-facing slopes (Coe et al., 2014; Ebel et al., 2015). The cohesion supplied by the roots is responsible for the connection observed between landslide distribution and slope aspect (McGuire et al., 2016). On the Loess Plateau, vegetation recovery is one of the main ecological measures for mitigating sediment loss (Fu et al., 2009). Increased soil strength and hydraulic conductivity due to strong root networks may enhance the topographic initiation conditions (Montgomery and Dietrich, 1994; Wang et al., 2020). North- and westward moving storms may potentially produce more intense rainfall on the south- and east-facing slopes. This assumption may be invalid if an aspect-dependent landslide distribution is present in a localized catchment with a specific vegetation community. This study highlights the effect of the mechanical function of plants on landslides. If an aspect-dependent landslide exists in a localized area with vegetation cover comprising the same plant species alongside a high level of vegetation cover, the aspect-dependent landslide initiation observed cannot be attributed to the mechanical effect of plant roots.

To determine the relationship observed among vegetation, landslides, and slope aspect, the effects of the physical properties and strength of hillslope materials cannot be excluded. On the northern part of the Loess Plateau, China, as well as in many other semi-arid environments, different types and densities of vegetation and soils develop on north-facing versus south-facing convergent slopes (Fu, 1983; Heimsath et al., 1997; Wang, 2008). This is because systematic differences in the amount of direct sunlight translate into differences in physical and chemical weathering. North-facing convergent slopes have lower evaporation rates, retain snow cover longer in spring, and tend to hold soil moisture longer during the summer growing season. These differences may result in localized ecosystem communities in the presence of trees or shrubs on grass. South-facing slopes experience heavier and more frequent hydration, thermal expansion, or freeze-thaw cycles due to day warming and night cooling and tend to have stronger weathering throughout the year. These differences can result in local differences in the grain component, soil strength, and soil profile. This has indirect effects at the landslide scale through the mechanics of excessive pore-water pressure dissipation and sliding surface liquefaction (Terzaghi, 1950; Sassa, 1984), and hillslope hydrology behavior (Godt et al., 2009; Lee and Kim, 2019). Therefore, the physical properties of hillslope materials may be attributed to the aspect-dependent landslide initiation observed.

Shallow landslides are examples of debris flow initiation, which often enlarges their scale by multiple mechanics (Hungr et al. 2005; Iverson et al. 2011). When the slope fails, the pore-water pressure abruptly increases within the shear zone (Iverson and LaHusen, 1989; Wang and Sassa, 2003). If the excessive pore-water pressure persists high over the static pressure for a relatively long duration, the displaced masses enlarge their volume by widespread liquefaction and transform into debris flows (Bogaard and Greco, 2016). The magnitude of the pore-water pressure is closely related to the scale of the shallow landslide. Therefore, the scale of shallow landslides can be determined by excessive pore-water pressure during the failure process. However, the aspect-dependent landslide distribution in these two areas refers to the differences in landslide probability rather than the landslide scale.

In the present study, we used a combination of field soil moisture observation, strength measurement, hydraulic conductivity analysis of hillslope materials, and numerical modeling of slope stability to explain the high potential for landslide initiation on south-facing slopes relative to north-facing slopes with the same vegetation communities. Differences in landslide geometry and initiation conditions, in the form of the contributing area above the scar area and the landslide gradient, were shown using field studies and high-resolution GeoEye-1 images. The differential weathering-related physical properties and strength of the soil mass, including the dry unit weights, porosity, grain

size, effective cohesion, and inner friction angle were examined. We have also highlighted the importance of excessive pore-water pressure, hillslope hydrology, and stability in explaining the aspect-dependent landslide initiation observed. The results of this work will deepen our understanding of aspect-dependent landslide distribution in some mountainous areas of the Northern Hemisphere.

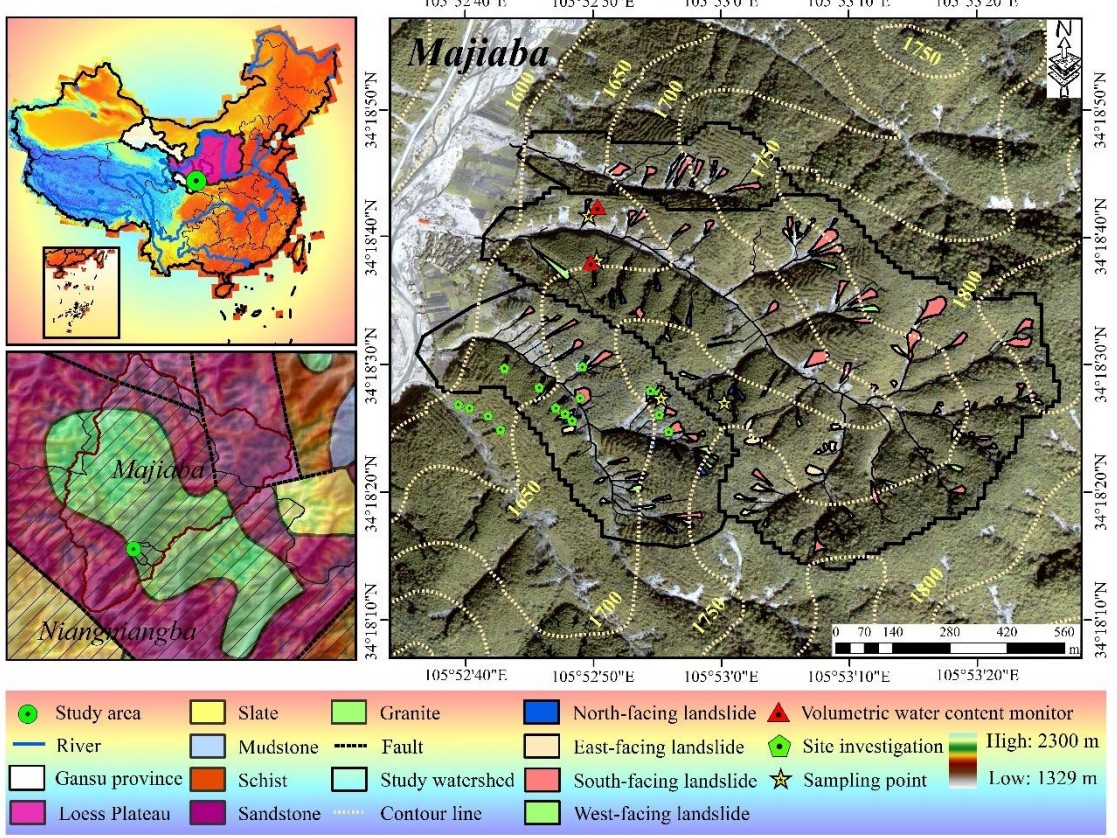

**Fig. 1.** Location, topography, and simplified lithology of the study area. All maps are created by the authors. The graph of Majiaba was taken using an Unmanned Aerial Vehicle. The territorial domain of China and simplified lithology map are from China Geological Survey. Elevation legend refers to the mountain spanning Niangnaingba and Majiaba.

## 2 Study area

The study area is in the mountainous region of Majiaba village, northeast of Niangniangba town, Tianshui City, Gansu Province, Central China. It is also close to the dividing crest of the Yellow and Yangtze Rivers and on the eastern part of the Loess Plateau. The elevation of the mountain near Niangniangba town in the mountain region of the study area ranges from 1329 m to 2300 m. Most of the hillslope is underlain by sandstone, and the stratigraphic units of granite, slate, schist and mudstone account for a smaller area. This area has four distinctive seasons and a semi-humid climate. The annual precipitation is approximately 491.6 mm and predominantly falls during June and August. One branch fault of the Tianshui-Lanzhou fault system runs through the area and has had no rupture records for the last few decades.

The shallow landslides in the study area and nearby surroundings were triggered by the prolonged antecedent precipitation during July 1–24 and the intensive rainstorm on July 25, 2013 (Yu et al., 2014; Guo et al., 2015). Most shallow landslides in the entire storm spanned the mountain area with a gradient of 20–25 °, located on south-facing slopes and in areas with relatively low-coverage vegetation (Li et al., 2021). Besides, some works found that plant roots may increase the topographical initiation threshold of landslides because of their positive effect on the strength

and hydraulic conductivity of soil-root composite (Dai et al., 2022). The three small catchment areas in the Majiaba Watershed are underlain by granite units. The total area is 0.88 km$^2$ with vegetation cover of over 90% (Fig. 1). The relative relief was approximately 200 m, and the mean hillslope gradient was 37°. The reason why the three catchments in the area were chosen is that the main plant species on the south- and north-facing slope is *Larix kaempferi*, which commonly have highly developed lateral roots with depth < 0.4 m. However, landslides in the three catchments still have a higher propensity for occurrence on south-facing slopes in comparison with the north-facing slopes. This finding differs from the results from Frontal Range, Colorado, USA, and the Central Loess Plateau, where landslides commonly occur in sparsely vegetated areas. Li et al. (2021) only addressed the relationship between landslide probability and vegetation cover at the regional scale, while excluding the importance of the properties of hillslope materials at a more localized scale. Therefore, we hypothesize that such observations in the study area may not be the result of the mechanical effect of plant roots but may be from the distinctive physical properties and strength of hillslope materials due to differential weathering.

## 3 Materials and methods

### 3.1 Landslide information interpretation

The high resolution GeoEye-1 image (0.5 m × 0.5 m) on October 8, 2013 was orthorectified and the landslide boundary was visually interpreted using ENVI 5.1 and e-Cognition 8. An unmanned aerial vehicle (UAV) was used to obtain a digital elevation model (DEM) with a 5 m resolution. The GeoEye-1 orthographic image and DEM were spatially registered in ArcGIS 10.2 as a standard layer of orthoimage. The landslide initiation condition is represented by the competition between the slope gradient and upslope contribution area ($A-S$):

$$S = kA^{-b} \tag{1}$$

where $S$ is the local slope (m/m); $A$ is the contribution area above the landslide head scar (m$^2$); $k$ is an empirical constant related to lithology, vegetation, and climate; and $b$ is an empirically defined index.

Field studies were conducted to measure the depth of the head scar and sidewall area using tape, and the failure depth was taken as their average. The landslide volume could then be calculated using the interpreted scar area and failure depth measured. Detailed landside information including the landslide number and area probability, landslide volume and width, head scar and sidewall depth, and the upslope contributing area–slope gradient condition for the south- and north-facing slopes were compared.

### 3.2 Field monitoring and soil sampling

To investigate the hillslope hydrology on south- and north-facing slopes, Frequency Domain Reflectometry (FDR) soil moisture sensors were used in this work to record the volumetric water content. To avoid the randomness of data caused by natural factors such as terrain and vegetation, a total of 16 shallow landslides were investigated to excavate soil profiles and take undisturbed soil samples. Sensors were installed at depths of 30 cm, 70 cm, and 110 cm on the south- and north-facing slopes to monitor the volumetric water content during June and September 2021. Soil moisture monitoring was implemented at two concave sites on the south- and north-facing slopes. The meteorological station was less than 3 km away from the study area to record the rainfall on a 30 min basis. During the sensor installation, undisturbed soil samples near the sensor location were taken for indoor tests, including the dry unit weight, porosity, grain size, shear strength, and hydraulic conductivity. The grain size was analyzed using a Malvern MS 3000 instrument (Malvern, England). In each layer, at least four samples were collected for the consolidated undrained triaxial compression test (CU). Two samples were collected for unsaturated hydraulic conductivity measurement using transient release and imbibition tests (Lu and Godt, 2013). Saturated hydraulic conductivity was determined using the constant water head method (Table 1).

### 3.3 Pore-water pressure dissipation

CU tests were performed to obtain the effective cohesion, effective internal friction angle, and pore-water

pressure curves. Soil samples with a diameter of 50 mm and height of 100 mm were first saturated in a vacuum pump. They were then consolidated in the chamber of the GDS apparatus at 50, 100, 150, and 200 kPa confining pressures and 10 kPa backpressure. During each test, the shearing rate was set to 0.1 mm/min, and the device automatically recorded data every 10 s. Owing to the varied particle components and soil texture, the increasing and dissipation ratios of pore-water pressure differentiate a lot. As a high excessive pore-water pressure and slow dissipation ratio could cause widespread Coulomb failure within the shear zone, it will influence the landslide scale. To compare the rate of rise and dissipation of pore-water pressure during the CU test, the ratio is expresses as

$$i = \frac{p_{t+\Delta t} - p_t}{\Delta t} \tag{2}$$

where $i$ is the increase or dissipation ratio of the excessive pore-water pressure, and $p_t$ and $p_{t+\Delta t}$ are the pore-water pressures measured during the time interval of $\Delta t$. A higher $i$ indicates that the pore-water within soil mass drainage rapidly and the pore-water pressure will dissipate in a short time. In other words, the $i$ is a proxy representing the hydraulic conductivity.

**3.3 Water storage and drainage**

The unsaturated permeability of soil mass (diameter 61.8 mm, height 25.4 mm) was measured using the Transient Release and Imbibition method (TRIM) (Lu and Godt, 2013). In this test, the water outflow mass was measured on a 10 min basis. In each test, air pressures of 250 kPa and 0 kPa corresponded to the drying and wetting processes, respectively. The Soil Water Characteristic Curve (SWCC) and Hydraulic Conductivity Function (HCF) were obtained using Hydrus 1-D (Wayllace and Lu, 2012). Using the models proposed by Mualem (1976) and van Genuchten (1980), the constitutive relations between the suction head ($h$), water content ($\theta$), and hydraulic conductivity ($K$) under drying and wetting states can be represented by the following equation:

$$\frac{\theta - \theta_r}{\theta_s - \theta_r} = \left[ \frac{1}{1 + (\alpha|h|)^n} \right]^{1 - \frac{1}{n}} \tag{3}$$

and

$$K = K_s \frac{\left\{ 1 - (\alpha|h|)^{n-1}[1 + (\alpha|h|)^n]^{\frac{1}{n} - 1} \right\}^2}{[1 + (\alpha|h|)^n]^{\frac{1}{2} - \frac{1}{2n}}} \tag{4}$$

where $\theta_r$ is the residual moisture content (%), $\theta_s$ is the saturated moisture content (%), $\alpha$ and $n$ are empirical fitting parameters, $\alpha$ is the inverse of the air-entry pressure head, $n$ is the pore size distribution parameter, and $K_s$ is the saturated hydraulic conductivity (cm/s).

The soil water storage ($S_s$) and drainage ($S_d$) during a rainfall event can be evaluated by the soil depth and the difference between the maximum soil moisture and antecedent soil moisture:

$$S_e = \frac{\theta - \theta_r}{\theta_s - \theta_r} \tag{5}$$

$$S_s = S_e^w \Delta h \tag{6}$$

$$S_d = P - S_e^d \Delta h \tag{7}$$

where $S_e$ is the degree of saturation, $\theta$ is the volumetric moisture content measured (%), $\Delta h$ is the average soil thickness (400 mm in this study), $S_e^w$ and $S_e^d$ are the residual soil moisture in the wetting and drying processes (%), and $P$ is the accumulated rainfall (mm).

**3.4 Stability fluctuation**

In this study, we applied a finite and infinite stability model to assess the slope stability fluctuation during the rainy season as an attempt to examine aspect-dependent landslide initiation from the perspective of classical mechanics and the state of stress (Schmidt et al., 2001). The finite-slope model evaluates the stability $F_s'$:

$$F_s' = \frac{c_l A_l + c_b A_b + A_b (\rho_s - \rho_\omega S_e) \, gz \cos^2 \beta \tan \varphi'}{A_b \rho_s gz \sin \beta \cos \beta} \tag{8}$$

where $\beta$ is the topographic slope angle (°), $A_l$ is the lateral area of side wall, m$^2$, $A_b$ is the basal area, m$^2$, z is the sliding depth (m), $c_l$ is the effective cohesion along the sidewall (kPa) and adopts the cohesion of layer 1 and layer 2, $c_b$ is the basal soil cohesion (kPa), and adopts the cohesion of layer 3, $\rho_s$ is the soil particle density, g/cm$^3$, and $\rho_w$ is the water density, g/cm$^3$.

The infinite slope stability model in this study provides insight into the stress variation resulting from changes in the soil suction and water content during infiltration (Lu and Likos, 2006):

$$F_s'' = \frac{\tan \varphi'}{\tan \beta} + \frac{2c'}{\gamma z \sin 2\beta} - \frac{\sigma^s}{\gamma z}(\tan \beta + \cot \beta) \tan \varphi' \tag{9}$$

where $\varphi'$ is the effective friction angle, °; $\beta$ is the topographic slope angle, °; $c'$ is the effective cohesion, kPa; $\gamma$ is the unit weight of the soil, KN/m$^3$; and $\sigma^s$ is the suction stress (kPa), expressed as:

$$\sigma^s = -\frac{S_e}{\alpha}\left(S_e^{n/(1-n)} - 1\right)^{1/n} \tag{10}$$

## 4 Results

### 4.1 Shallow landslides on south- and north-facing slope

In the study area, the direct sunlight does not coincide with the aspect orientation because it is in the north the Tropic of Cancer. The south-facing slope is defined between 157.5 ° and 247.5 ° and the north-facing slope is between 0 ° to 67.5 °, and 292.5 ° to 360 ° (0 ° is the due north). There were 71 shallow landslides on the south-facing slope and 20 landslides on the north-facing slope in the study area. Figure 2a shows that shallow landslides on south-facing slopes have larger spatial areas than those on north-facing slopes. Most of the shallow landslides occurred on the south-facing slope (Fig. 2b). The volume of landslides on the south-facing slope was greater than that on the north-facing slope. For landslides on the south-facing slope, the basal area was 372.64 m$^2$ and the width was 14.9 m on average. For landslides on the north-facing slope, the average basal area was 157.28 m$^2$ and the width was 7.7 m (Fig. 2c). Although the landslides on the south-facing slope had a larger volume and greater width, the depth of the head-scar and sidewall area are no greater than those on the north-facing slope. Field studies showed that the averaged depth for landslides on the north-facing slope was 1.02 m, which was deeper than the depth of 0.83 m for landslides on south-facing slope (Fig. 2d). The landslides on the south-facing slope exhibited an overwhelming propensity for occurrence in terms of number and area. Meanwhile, the failure depth was no more than that of the landslides on the north-facing slope.

Shallow landslides can be modeled as occurring when sufficient through-flow converges from the upslope contribution area to the hollow area and triggers slope instability (Montgomery and Dietrich, 1994). Their topographic initiation conditions are controlled by the spatial competition between the slope and upslope contribution being area dependent (Stock and Dietrich 2003 and 2006; Horton et al., 2008). For the shallow landslides in the study area, the averaged upslope contributing area and slope gradient did not significantly differ (Fig. 3a). Meanwhile, the lower limit line representing the minimum initiation condition for landslides on south-facing slopes was lower than that on the north-facing slopes (Fig. 3b). This indicates that a higher upslope contribution area was required to provide sufficient through-flow conditions and trigger slope failures on the north-facing slope. Given that the landslides in the study area were triggered by prolonged antecedent precipitation and intensive rainfall (Li et al., 2021), sufficient rainfall infiltration could result in a high soil water content within the displaced mass, leading to a decrease in matric suction and soil strength. The generation of pore-water pressure in response to intense rainfall also plays an important role in shallow landslides. Therefore, we have proposed two assumptions to elucidate the distribution and scale of aspect-dependent landslides. The first assumption is that the basal area of the landslide may be related to the soil strength and high pore-water pressure. This assumption can be tested by the pore-water properties, including the pore-water generation potential and dissipation ratio during the failure process. The second

assumption is that the south-facing slope may have a higher failure potential than the north-facing slope in given
rainfall process. This can be determined from the stability comparison using equations (8) and (9).

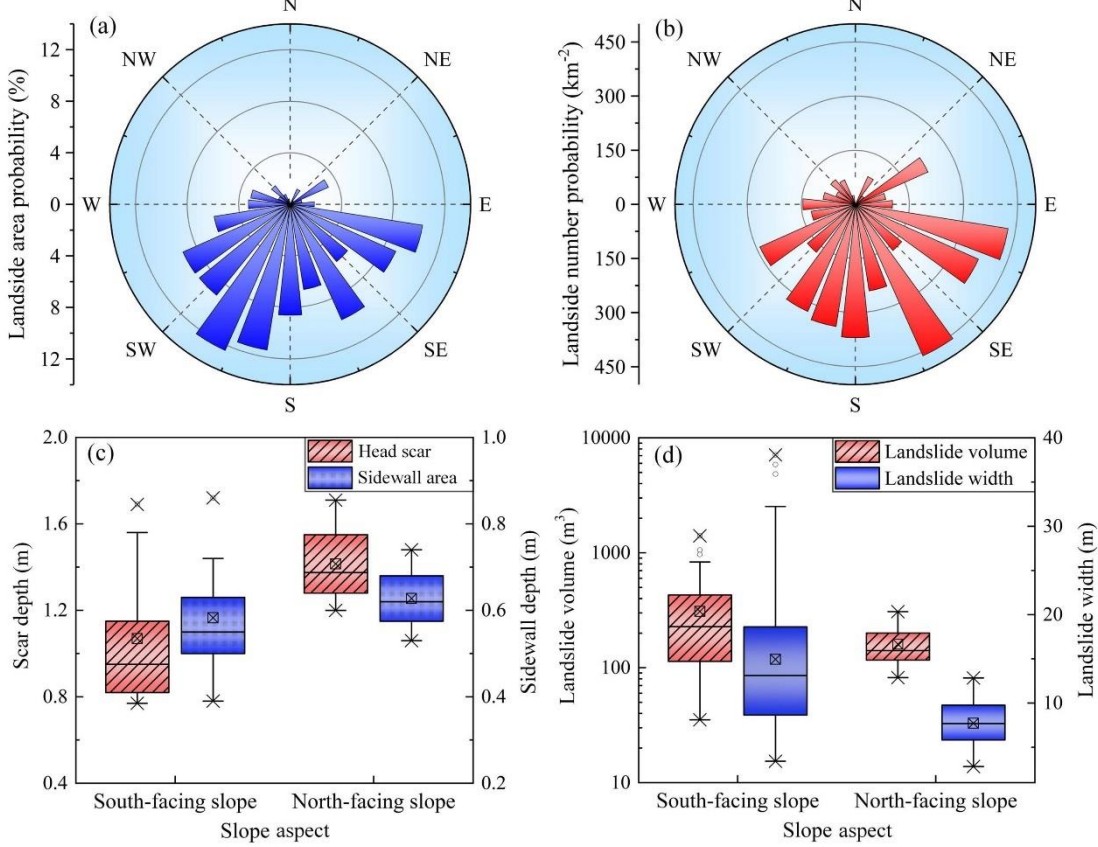

**Fig. 2.** Spatial distribution and geometric characteristics of the landslide: (a) Landslide area probability vs slope
aspect; (b) landslide number probability vs slope aspect; (c) landslide volume and width vs slope aspect; (d)
scar depth and sidewall depth vs slope aspect. The edge line of box in the box chart shows the 75th quantile
(Q3), median and 25th quantile (Q1) from top to bottom. The length of the box is referred to as the inter-quartile
range (IQR). The crossed square inside the box is the average value. The whiskers extend to the maximum and
minimum values except the mild outliers. The upper limit and lower limit of whiskers are Q3+1.5IQR and Q1-
1.5IQR respectively. The circles are the outliers, and the cross symbol is the maximum and minimum values
for all the data.

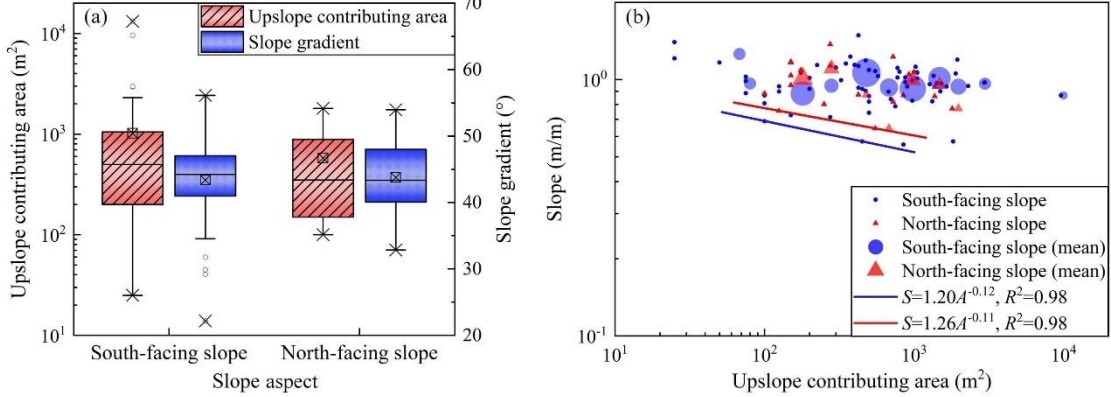

**Fig. 3.** Upslope contributing area and slope gradient condition: (a) Upslope contribution area and mean slope vs
slope aspect; and (b) the upslope contributing area vs mean slope gradient above the landslide area. The
definitions of the whiskers are shown in caption of fig. 2. The circles are averaged slopes with the radius size

proportional to the number of landslides. The small cross represent all individual data values. The power-law
regression is fitted with the dataset closet to the axis origin.

## 4.2 Differences in soil physical properties

To show the differences in the physical properties of the hillslope materials, the dry unit weights, porosity, and
grain size distribution of the soil mass in the three layers on each slope were compared (Fig. 4). The effective
cohesion and inner friction angle were then examined with respect to the particle component (Table 1 and Fig. 5).

Table 1 Physical properties and strength parameters of the soil mass

| Parameters | South-facing slope | | | North-facing slope | | |
|---|---|---|---|---|---|---|
| | Layer 1 | Layer 2 | Layer 3 | Layer 1 | Layer 2 | Layer 3 |
| Unit weight of soil (kN/m$^3$) | 14.8 | 15.6 | 17.2 | 14 | 16.6 | 17.1 |
| Porosity (%) | 43.0 | 43.1 | 36.2 | 42.5 | 37.3 | 36.4 |
| Effective cohesion (kPa) | 6.5 | 17.5 | 21.2 | 5.3 | 9.1 | 7.9 |
| Effective inner friction angle (°) | 29.8 | 25 | 31 | 27.1 | 35.2 | 41 |
| Saturated hydraulic conductivity (cm/s) | $6.4 \times 10^{-3}$ | $6.2 \times 10^{-4}$ | $4.4 \times 10^{-4}$ | $8.8 \times 10^{-3}$ | $1.2 \times 10^{-3}$ | $4.3 \times 10^{-3}$ |

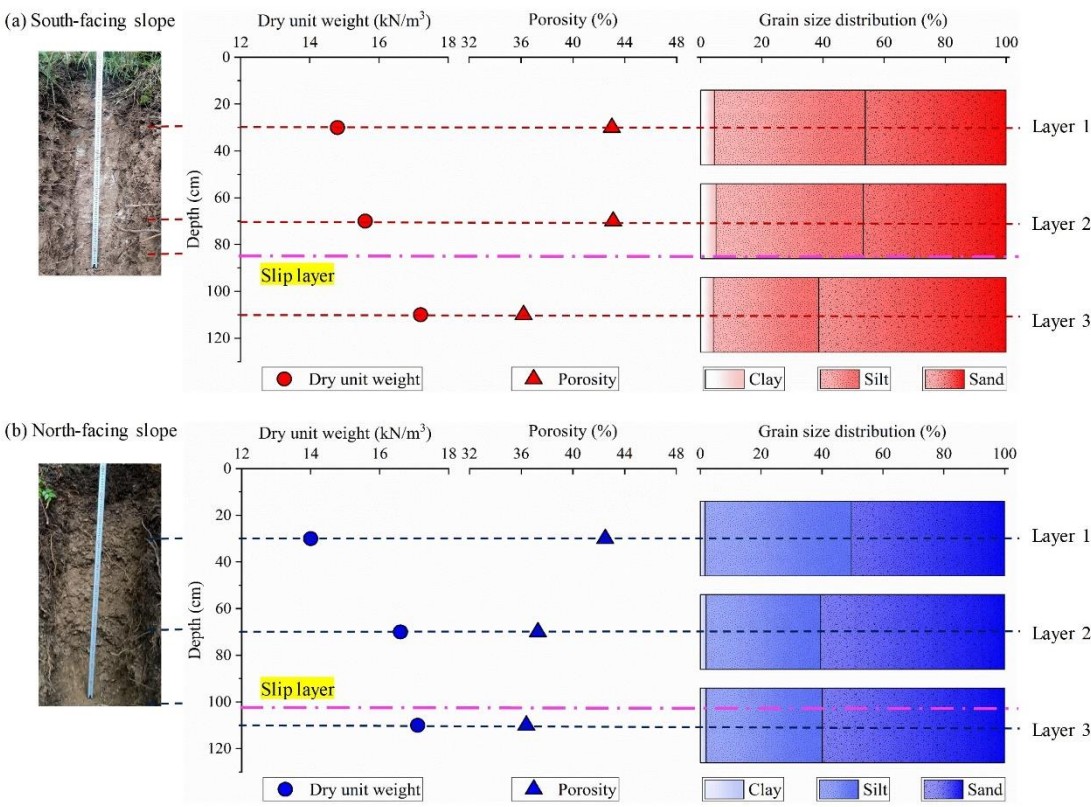

**Fig. 4.** Differences in the soil properties including dry unit weights, porosity, and grain size in sand, silt, and clay.
(a) Physical properties of soil mass on the south-facing slope; and (b) physical properties of soil mass on the
north-facing slope. The two-soil profile photos were taken by Yanglin Guo during field studies.

For the soil mass on the south-facing slope, the dry unit weights increased with soil depth, whereas the porosity
and saturated hydraulic conductivity decreased (Fig. 4a and Table 1). For Soil layers 1 and 2, the soil textures were
similar, because the proportions of sand, silt, and clay did not differ significantly. However, the proportion of silt in

Soil layer No. 3 was no more than that in layers No. 1 and 2, and the sand proportion was higher. The average failure depth was above Soil Layer No. 3 and below Soil Layer No. 2. For the soil mass on the north-facing slope, the dry unit weight also increased with soil depth. Unlike the south-facing slope, the porosity of the soil mass for the three soil layers was approximately 38% and did not differ among them. For the soil texture, the proportion of sand in Soil Layer No. 1 was no more than that in Soil Layers No. 2 and 3 (Fig. 4b). The depth of the failure plane was close to that of Soil Layer 3.

In comparison, one of the main difference was the higher saturated hydraulic conductivity for the soil mass above the failure plane on the north-facing slope. This may have resulted from the high porosity and sand proportion. This indicates that the rainfall infiltration on the north-facing slope could penetrate faster than that of the south-facing slope. The soil mass of the three layers on the south-facing slope had a higher proportion of fine particles than those on the north-facing slope if gravel was considered (Fig. 5). The saturated hydraulic conductivity for the soil masses from Soil Layers No. 2 and 3 on the south-facing slope was lower than that on the north-facing slope. This is expected because the porosity and proportion of fines on the south-facing slope were higher.

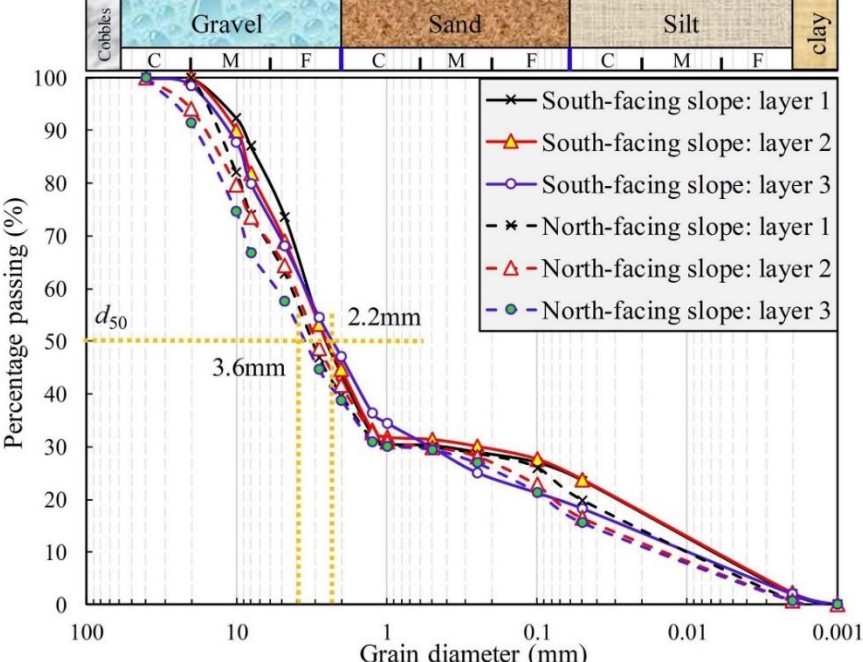

**Fig. 5.** Soil particle component curves

According to the results of the triaxial shear test (Table 1), the soil mass in each layer on the north-facing slope had a smaller effective cohesion than that on the south-facing slope. The effective cohesion on the failure plane for landslides on the south-facing slope may be twice that on the north-facing slope. However, the effective inner friction angles for the soil masses of Soil Layers 2 and 3 on the north-facing slope were far greater than those on the south-facing slope. These differences in effective cohesion and inner frictional angle may be attributed to the higher clay and silt and fewer coarse grains within the soil mass on the south-facing slope.

**4.3 Pore-water pressure properties**

The consolidation module of the triaxial shear test was used to measure the generation and dissipation process of the pore-water pressure. The principle is to consolidate and drain soil from the initial saturated state. Under the same confining pressure, there are pronounced differences in the consolidation rate, consolidation time, and peak rise in pore-water pressure for different soil properties. The results of the pore-water pressure during the consolidation process under 200 kPa effective confining pressure were compared here (Fig. 6). The peak value of pore-water pressure within the soil mass on the south-facing slope was higher than that on the north-facing slope.

The peak value of the pore-water pressure within the soil mass on the south-facing slope increased to 150–200 kPa.

However, the peak value of pore-water pressure within the soil mass on the north-facing slope was below 150 kPa.

Both the rising and decaying rates of pore-water pressure for Soil Mass layers 1 and 2 on the south-facing slope were

lower than those on the north-facing slope. The rate and decaying rates for Soil Mass layer No. 2 on the south-facing

slope were 1.2 kPa/10 s and −0.031 kPa/10 s, respectively. However, they were 9.6 kPa/10 s and −0.765 kPa/10 s

for the soil mass on the north-facing slope.

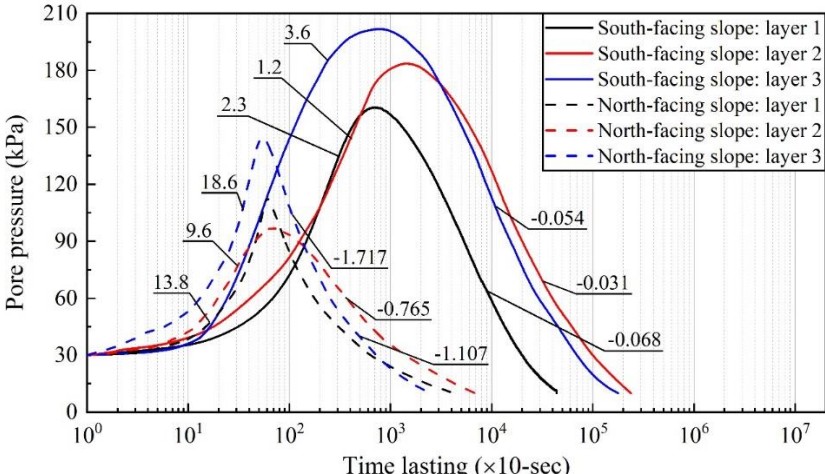

**Fig. 6.** Variation in pore-water pressure under effective confining pressure of 200 kPa by GDS triaxial shear

tests. The values in the figure 6 are the average rates of rise and dissipation of pore-water pressure during

consolidation calculated by Equation 2. The unit of *x*-axis marks the time record interval of 10 seconds.

The lower peak pore-water pressure demonstrates the effect of fine particles on the pore-water pressure, which

directly affects landslide mobility and scale. Rainfall-induced landslides result from an increase in positive pore-

water pressure within the failure plane, which reduces the effective stress and shear strength of the soil (Terzaghi,

1950). This often occurs in the undrained soil layer, which can easily cause slope liquefaction (Sassa, 1984). The

increase in pore-water pressure predominantly depends on the speed of landslide movement, soil deformation, and

soil permeability. If the shear rate is given, the dissipation rate of pore-water pressure for high-permeability soil is

faster, and therefore, the increase in pore pressure is smaller (Iverson and LaHusen, 1989; Iverson et al., 1997). As

shown in Table 1, the saturated hydraulic conductivity for soil mass of Layers No. 2 and 3 on the north-facing slope

was 10 times that of the south-facing slope. Therefore, the peak pore-water pressure measured during the test for the

soil mass on the south-facing slope was higher. The soil mass on the north-facing slope had higher sand and gravel

contents than that on the south-facing slope (Fig. 5). A high clay content on the south-facing slope filled the

macropores within the soil mass and reduced the pore-water discharge rate. Wang and Sassa (2003) found that fine

particles play the most important role in the dissipation of pore pressure. The pore-water pressure within the saturated

sand increased with shear rate. The soil mass with high coarse particles produced less pore water pressure than the

soil with high fine particles during the shear process. Therefore, the high permeability of the soil mass on the north-

facing slope may result in low peak-pore water pressure. The higher fine particles may result in a slow increase and

dissipation of the pore-water pressure. This slow pore-water pressure dissipation could result in the liquefaction

failure of the sliding mass and a larger landslide area.

**4.4 Unsaturated hydraulic conductivity**

4.4.1 Measured water outflow mass

Figure 7 shows the water outflow mass measured for a given 10 min period during the drying and wetting

processes. The water outflow masses measured for Soil Layers 2 and 3 on the north-facing slope were generally

higher than those on the south-facing slope. For the drying tests using the soil mass of Soil Layers No. 2 and 3 on the north-facing slope, the given water outflow masses were 0.102 g/10 min and 0.131 g/10 min, respectively. However, the water outflow masses measured for the soil mass of Soil Layers No. 2 and 3 were 0.077 g/10 min and 0.050 g/10 min, respectively, on the south-facing slope (Fig. 7a). For tests using the same layers of the soil mass in the wetting process, the water outflow masses measured were 0.051 g/10 min and 0.094 g/10 min on the north-facing slope, respectively, and 0.032 g/10 min and 0.027 g/10 min, respectively, on the south-facing slope (Fig. 7b). Overall, the permeability of the soil mass on the north-facing slope was higher than that on the south-facing slope. The same results were obtained when the saturated hydraulic conductivities of the soil layers were measured using the constant water head method (Table 1).

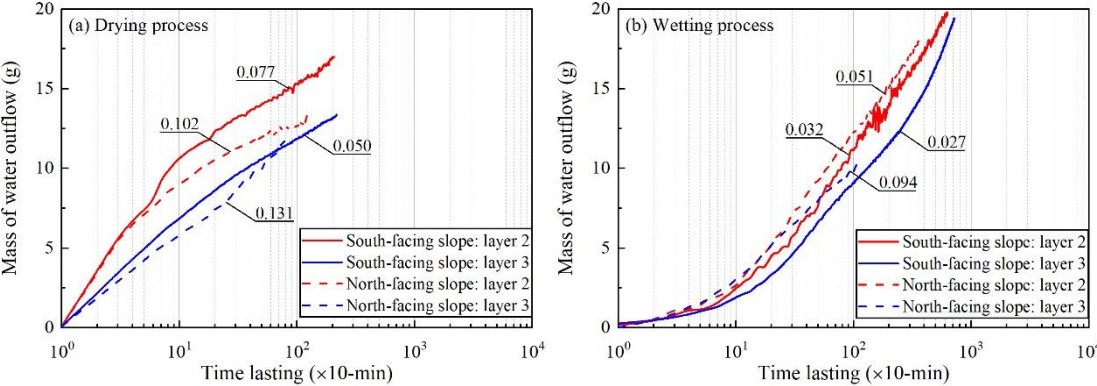

**Fig. 7.** Mass of water outflow during the drying and wetting process: (a) drying tests, (b) wetting tests. The software automatically records the mass of water outflow 10 min each, so the *x*-axis starts from $10^0$.

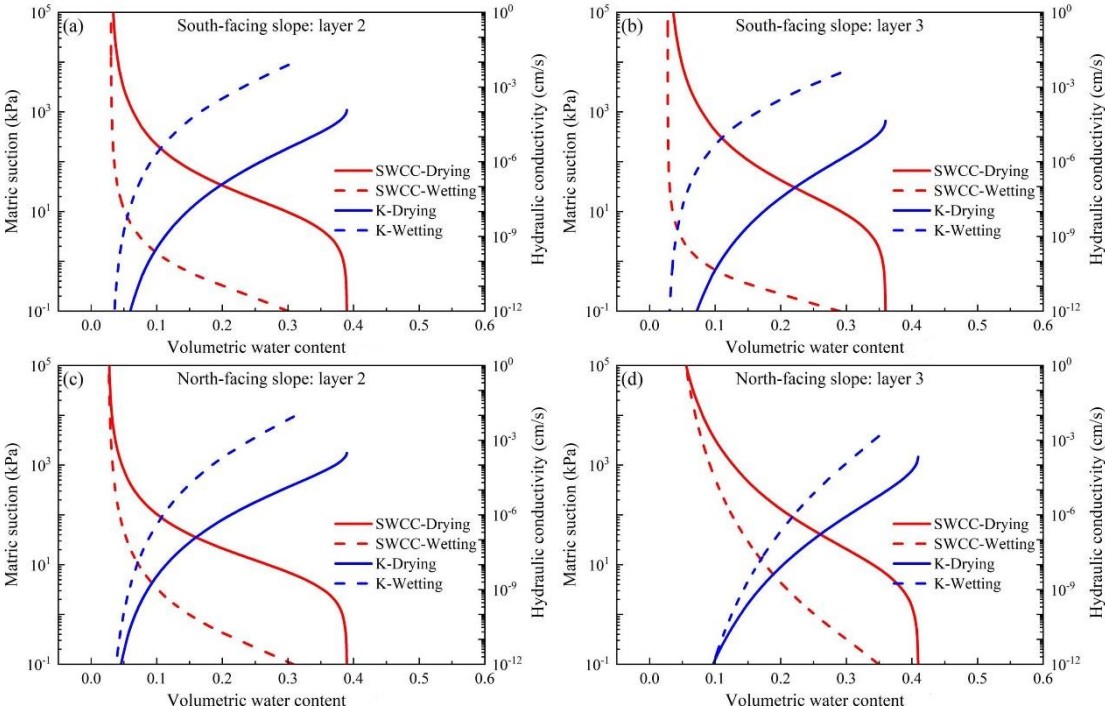

**Fig. 8.** Soil water curve obtained using the TRIM test: (a) Layer No. 2 on the south-facing slope, (b) Layer No. 3 on the south-facing slope, (c) Layer No. 2 on the north-facing slope, and (d) Layer No. 3 on the north-facing slope.

4.4.2 SWCC and HCF curves

The Soil Water Characteristic Curve (SWCC) and Hydraulic Conductivity Function (HCF) are critical for the analysis of water flow movement and mechanical behavior of unsaturated soil material. In this study, the Transient

Release and Imbibition Method (TRIM) for unsaturated hydraulic property measurement (Lu and Godt, 2013). The advantage of the TRIM method is that it combines physical experiments and calibration. It employs a relatively simple and reliable measurement of transient water content using an electronic balance to record the signature of transient unsaturated flow. It also takes advantage of the robust inverse modeling capability to simulate the physical process. The apparatus could accommodate both undisturbed and remolded samples. The results of this study were obtained using the Hydrus-1D code with the reverse modeling option, and the Levenberg–Marquardt non-linear optimization algorithm. This minimized the error between the results of the test and the simulation (Wayllace and Lu, 2012). Meanwhile, to ensure the uniqueness of the parameters, the algorithm repeatedly runs with different initial parameter estimates until it converges to obtain the same or similar results. The prediction results are then compared with the function curves of water flow and time obtained from the actual experiment so that they can be combined to meet certain accuracy requirements. In this experiment, the R square of the regression between the optimized predicted value and the observed value was greater than 0.99. The model constraint effect of the TRIM under two suction increment steps was better, and the parameters obtained by the inversion calculation were more accurate (Lu and Godt, 2013). Table 2 shows the soil parameters obtained using the Hydrus 1-D inversion.

Table 2 Parameters describing the Soil and Water Characteristic Curve (SWCC) and the Hydraulic Conductivity Function (HCF) from Hydrus 1-D

| Parameters | Definition | South-facing slope | | North-facing slope | |
|---|---|---|---|---|---|
| | | Layer 2 | Layer 3 | Layer 2 | Layer 3 |
| $\theta r$ | Residual moisture | 0.0302 | 0.0278 | 0.0262 | 0.0268 |
| $\theta s^d$ | Saturated moisture | 0.39 | 0.36 | 0.39 | 0.41 |
| $\theta s^w$ | | 0.36 | 0.38 | 0.39 | 0.42 |
| $\alpha^d$ (kPa$^{-1}$) | The inverse of the air-entry pressure head | 0.0128 | 0.0117 | 0.0156 | 0.0141 |
| $\alpha^w$ (kPa$^{-1}$) | | 0.78 | 0.94 | 1.21 | 1.86 |
| $n^d$ | The pore size distribution parameter | 1.49 | 1.39 | 1.57 | 1.27 |
| $n^w$ | | 1.63 | 1.85 | 1.43 | 1.18 |
| $K_s^d$ (cm/s) | Saturated hydraulic conductivity | $1.52\times10^{-4}$ | $0.64\times10^{-4}$ | $3.76\times10^{-4}$ | $4.56\times10^{-4}$ |
| $K_s^w$ (cm/s) | | $9.58\times10^{-2}$ | $4.93\times10^{-2}$ | $4.10\times10^{-1}$ | $4.68\times10^{-1}$ |

Notes: the superscript $d$ and $w$ indicate drying and wetting states.

Using these parameters, the SWCC and HCF curves of the soil mass at Soil Layers 2 and 3 on the north- and south-facing slopes can be drawn (Fig. 8). Air-entry pressure and residual water content are two important parameters that describe the hydrological and mechanical characteristics of the hillslope materials. The air-entry pressure represents the critical value at which air enters the saturated soil and starts to drain. For Soil Layer No. 2, the difference between the air entry values of the north- and south-facing slopes can reach 14.03 kPa (Figs. 8a and 8c). The residual water content and air-entry pressure of the south-facing slope were higher than those of the north-facing slope. For Soil Layer No. 3, the soil mass on the north-facing slope has the smallest air-entry pressure, which is 0.51 times that of the air-entry pressure of the south-facing slope (Figs. 8b and 8d). The saturated hydraulic conductivities of Soil Layers No. 2 and 3 on the south-facing slope were lower than those on the north-facing slope in both the drying and wetting processes. The saturated hydraulic conductivity of the soil mass on the north-facing slope in the wetting test was one order of magnitude higher than that on the south-facing slope. In Table 1, the saturated permeability coefficient measured by the constant head test method also shows that the soil mass on north-facing

slope has higher permeability. These results suggest that it is more difficult for the soil mass on south-facing slope

to absorb and drain water than the soil mass on the north-facing slope.

**4.5 Water storage and drainage**

To show the water storage during the rainfall process and the water drainage after the rainfall, the timely

recorded soil moisture at various soil layers and the rainfall process during June 11 and August 20 were used (Figs.

9a and 9b). In comparison, this is likely the most important finding, as it shows that the soil becomes nearly saturated

on the south slope, but not on the north slope. This implies that the soil water on the south-facing slope has difficulty

in draining water because of the presence of more fine grains and slow pore-water pressure dissipation. The stable

soil moisture from Soil Layers No. 2 and 3 for both slopes may be attributed to the long dry seasons in the study

area. The daily rainfall amount > 30 mm on July 9 and 23 resulted in an increase in soil moisture for all the slope

layers.

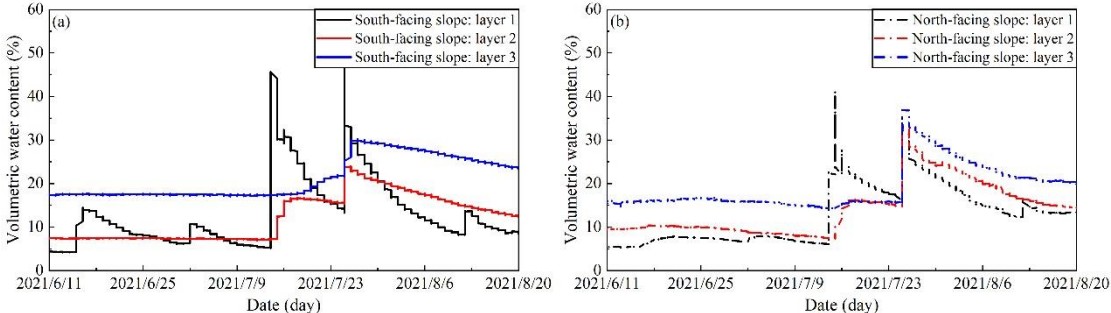

**Fig. 9.** Field monitored volumetric water content: (a) Soil moisture on the south-facing slope, and (b) soil moisture

on the north-facing slope.

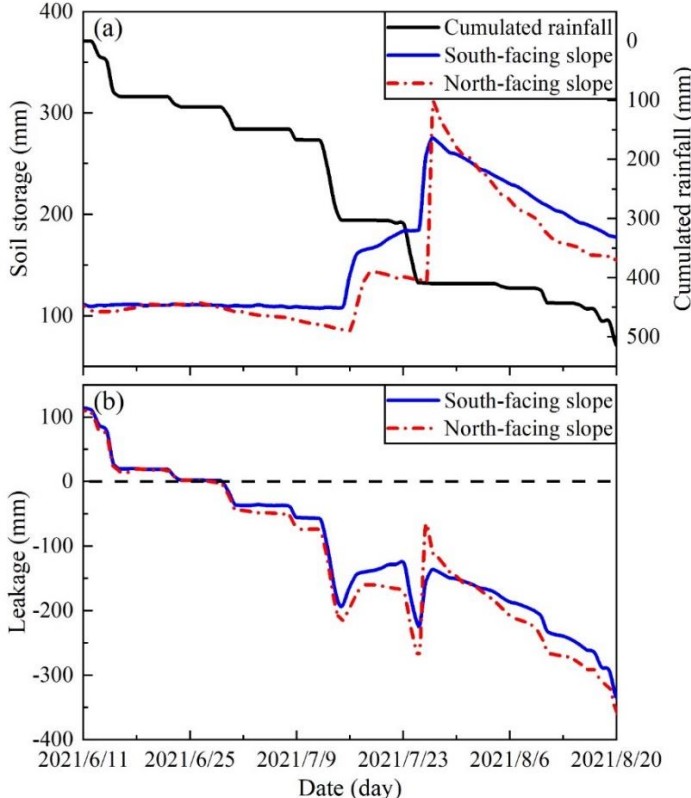

**Fig. 10.** Seepage model of slope water storage and drainage. (a) soil water storage, (b) soil water drainage

Figure10a shows that the storied water of the north- and south-facing slopes did not synchronously increase

with accumulated precipitation. When the storied water rapidly increased, the increase in soil water storage of the

north-facing slope was greater than that of the south-facing slope. On July 26, a rainfall of 30.8 mm/h was recorded, and the water storage of the slope reached the peak. The peak of the water storage on the north-facing slope was higher than that of the south-facing slope. However, when the accumulated rainfall tends to be stable, that is, when the rainfall stops for a period, the decline rate of the soil water storage on the north-facing slope is substantially higher than that on the south-facing slope. The soil water storage of the south-facing slope was always higher than that of the north-facing slope during rainfall. During the drainage process, the seepage rate of the north-facing slope was greater than that of the south-facing slope (Fig. 10b). Therefore, the south-facing slope had a better water storage performance, and the north-facing slope had a higher drainage performance.

**4.6 Stability fluctuation**

In this study, the infinite slope model and the finite slope model were used to characterize the sensitivity of landslide triggering to determine the main mechanism of high landslide probability on south-facing slopes. The infinite slope model can be used to examine the transient stress changes caused by water entering the soil, emphasizing the differences in soil permeability (Lu and Likos, 2006; Lu and Godt, 2013). The finite slope model focuses on the cohesion of the base surface and lateral periphery of the ground landslide source body, as well as the influence of the additional lateral cohesion provided by the vegetation root system for the landslide (Schmidt et al., 2001; Dai et al., 2022).

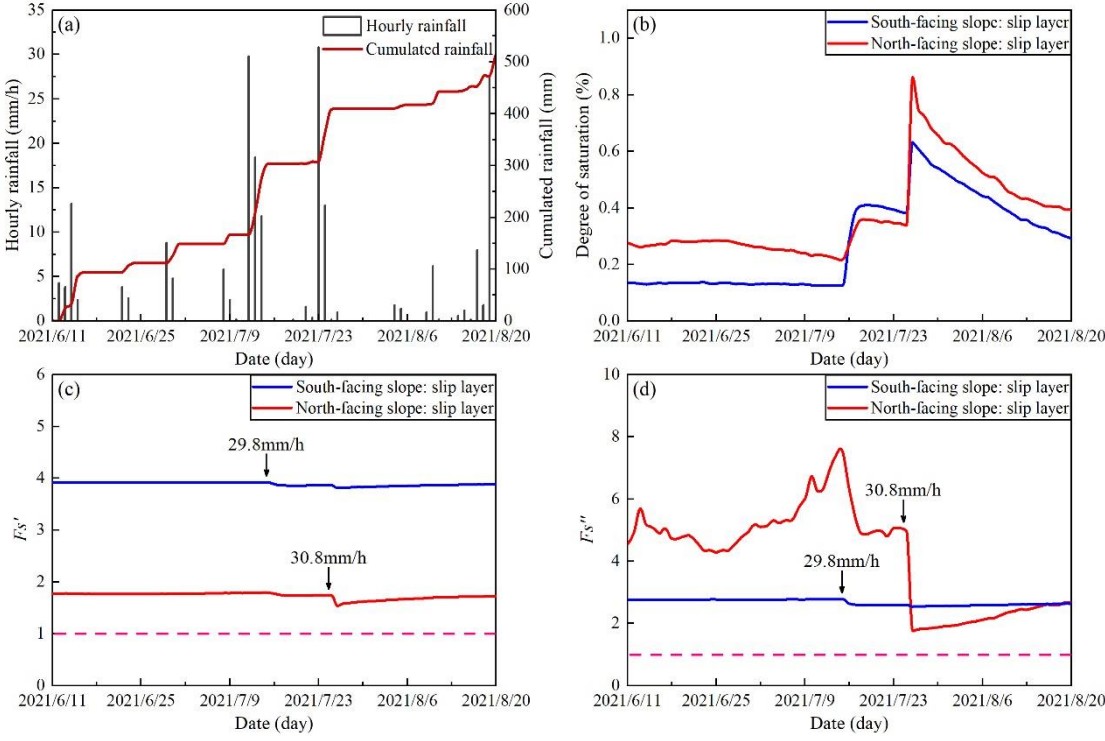

**Fig. 11.** Change in slope stability fluctuation: (a) rainfall records, (b) degree of saturation, (c) stability of finite slope model, and (d) stability of infinite slope model. The pink dotted lines indicate the stability index equals to 1.0.

Figure 11a shows the rainfall records from June 11 to August 20, 2021. In general, the degree of saturation of the sliding layer on the south-facing slope was higher than that on the north-facing slope (Fig. 11b). In the finite model, the stability of the south-facing slope was always higher than that of the north-facing slope (Fig. 11c). In the infinite model, the stability of the north-facing slope was generally higher than that of the south-facing slope, and the stability of the north-facing slope fluctuated substantially (Fig. 11d). On July 26, a rainfall event with a maximum intensity of 30.8 mm/h resulted in a sudden decrease in stability. The estimated stability index of the north-facing slope decreased to become lower than that of the south-facing slope and then increased afterwards. Although the soil

moisture of the south-facing slope increased substantially during the rainfall event on July 16, the stability fluctuation was relatively small. This may be related to the relatively strong effective cohesion and smaller pore structure. In finite slope model, the results have shown that the south-facing slope has a relatively high stability. However, this result contradicts to the high landslide density on the south-facing slope in the study area. In fact, the finite slope model does not consider suction stress, and the effective cohesion of hillslope materials mainly affects the stability result. In contrast, the results of the infinite slope model asserts that the state of the stress of the soil or regolith is modified by infiltration and changes in soil matrix suction. Furthermore, the fluctuation in fig. 11d also proves that the role of infiltration of water into shallow soils and the subsequent pore-water pressure response at depth is critical to the understanding the transient conditions that lead to shallow slope failure, because the stability fluctuation amplitude of the south-facing hillslope was smaller than that of the north-facing hillslope. This indicated that the water movement on the south-facing slope was less active than that of the north-facing slope. Therefore, in the study area, the change in soil suction stress was more sensitive to slope stability than the change in root soil cohesion. The change in soil permeability caused by differential weathering of the bedrock could be responsible for aspect-dependent landslide initiation in the study area.

**5 Discussion**

The strong propensity for landslides in some arid environments in the Northern Hemisphere is scientifically interesting, and some researchers have highlighted the contribution of plant roots. This finding is to be expected in the future in other mountain regions, where water is a limiting factor for local system sustainability. In the Colorado Frontal range, McGuire et al. (2016) found that the apparent cohesion supplied by roots was responsible for the connection observed between landslide distribution and slope aspect (Ebel, 2015; Rengers et al., 2016). In the study area, Li et al. (2021) also found that plant roots may explain the connection observed between vegetation cover and landslide probability for the entire study area. Dai et al. (2022) found that a strong root network and high saturated hydraulic conductivity may promote the $A-S$ condition of shallow landslides. On the Loess Plateau in China, some researchers have observed that the strong propensity for shallow landslide initiation is closely related to the present-day tree density, and plant roots do not penetrate over the failure plane (Guo et al., 2020; Deng et al., 2022). However, the strong propensity for shallow landslides on north- and south-facing slopes cannot be attributed to plant roots, because the artificial vegetation on both slopes is the same. Conversely, these observations could be the result of the soil hydraulic and mechanical properties from differential weathering.

This study has contributed to knowledge of the effect of differential weathering on aspect-dependent landslide initiation from the perspective of soil hydraulic properties, in addition to the mechanical and hydrological effects of plant roots. Except for the strong propensity for a high number of landslides, shallow landslides on south-facing slopes have exhibited larger areas and greater widths than those on the north-facing slopes (Fig. 2). This may be attributed to the fact that the slow dissipation of excessive pore-water pressure because widespread liquefaction may cause extend the landslide scale. For the thinner slip layer of landslides on south-facing slope, it may result from differential weathering, because the theoretical maximum or maximum slip layer for strong-cohesive slope should be larger than weak-cohesive slope at given slope (Iida, 1999; D'Odorico and Fagherazzi, 2003). One of the reasons may be that cohesive soil mass often hold tight together to displace downslope owing to the strength loss. The relatively weak-cohesive soil mass often loosens to displace downslope, with the slip layer close to the boundary between soil mass and bedrock underneath. However, a stronger effective cohesion tends to promote the $A-S$ conditions of shallow landslides. A larger up-slope contributing area or steeper gradient is required to trigger slope failure. Figure 3 shows that some shallow landslides on south-facing slopes fail at lower upslope contributing areas. Therefore, soil hydraulic property-related factors, such as the rising or dissipation of pore-water pressure, water storage, and drainage, may contribute to the phenomena observed.

The saturated hydraulic conductivities obtained by the constant water head and TRIM methods coincide, which demonstrates that the hillslope material on the north-facing slope has a larger water infiltration (Tables 1 and 2). However, the difference between $K_s^d$ and $K_s^w$ is strikingly high and the $K_s^d$ is smaller. Although the Trim test in this work measures the permeability of soil matrix, the influence of other factors, such as the soil development and weathering, preferential flow pathway and macro pore, cannot be ignored (Lohse and Dietrich, 2005; Maier et al., 2020) , and the contribution of such influence on the permeability rate cannot be evaluated at present. The results of the stability analysis using the finite and infinite models imply that the failure potential of slides on a north-facing slope is lower than that on a south-facing slope, because the stability index of south-facing slope is always close to 1.0. These differences imply that slope failures on a north-facing slope may only occur under intensive rainfall conditions or by a combination of prolonged antecedent precipitation and short duration intensive rainfall. For potential failures on south-facing slopes, the combination of prolonged antecedent precipitation and short duration intensive rainfall should be a potential trigger owing to the low hydraulic conductivity and pore-water pressure dissipation. This study highlights the role of hydraulic properties in landslide occurrence. Although the south- and north-facing slopes are underlain by granite, the physical properties of hillslope materials such as excessive pore-water pressure, strength of sliding mass, soil water storage, and leakage are significantly different. One of the possible limitations of this work lies in that the representativeness of the moisture observation and the uncertainty. Considering the multiple factors influencing landslides, the study area is selected with same bedrock underneath and similar plant species. Then, the moisture observation sites were selected on condition that similar soil profile, landscape with majority of landslides and the common topographical conditions. Therefore, this finding cannot be random because the study area has been selected on the condition that it is relatively far from the northern and eastern areas where local soils are predominantly loess deposits, and the study areas of Li et al. (2021) and Dai (2022), where the bedrock underneath differs substantially. The main purpose of this work is to elucidate the reason for aspect-dependent landslide initiation from the perspective of soil hydraulic properties. These differences result from differential weathering owing to the amount of direct sunlight. Other methods such as numerical or relative dating methods and preferential flow in the macropore distribution could provide new evidence for such observations.

**6 Conclusion**

Previous research on the strong propensity for shallow landslides on south-facing slopes over north-facing slopes has highlighted the role of plant roots. In a localized area with the same vegetation including plant roots, they do not penetrate the failure layer. Such overwhelming landslide phenomenon cannot be attributed to plant roots and may result from the differential weathering of bedrock under the influence of hydrothermal conditions. In this study, we jointly explained the soil hydraulic properties from physical and mechanical properties, pore-water pressure, unsaturated hydraulic conductivity, water storage and drainage, and slope stability fluctuation during monitoring, and studied landslide initiation related to slope direction. The following conclusions were drawn:

(1) In terms of soil physical and mechanical properties on both slopes, the soil masses on the south-facing slope have higher silt content than those on the north-facing slope. The effective cohesion of the soil mass on the south-facing slope was higher than that on the north-facing slope, while the effective frictional angle was smaller.

(2) The results of the GDS tests showed that the dissipation rate of pore-water pressure for soil mass on the south-facing slope was substantially lower than that on the north-facing slope. Higher effective cohesion and slower pore-water pressure dissipation may result in a larger basal area for shallow landslides on south-facing slopes.

(3) The soil mass on the south-facing slope had a higher residual water content and air entry pressure, and a lower saturated hydraulic conductivity than that of the north-facing slope. For water storage and drainage performance, the storied water from the south-facing slope was higher than that of the north-facing slope, while the north-facing slope had a higher leakage rate. The results of the stability analysis based on the finite and infinite

models show that the infinite slope model may be suitable for elucidating aspect-dependent landslide distribution in the study area.

**Acknowledgements**

This study was supported by the State Key Program of National Natural Science of China (Grant No. 42130701), the National Nature Science Foundation of China (42177309), and the Fundamental Research Funds for the Central Universities (Grant No. 2018BLCB03). The authors sincerely thank the contributions of other colleges, including Muyang Li, Zhisheng Dai, Lv Miao, Lijuan Wang, and Jiayong Deng, for their previous work near the study area.

**Code/Data availability**

The raw/processed data in this work cannot be shared at this time, because the data also form part of an ongoing study.

**Author contributions**

Professor Ma Chao found a strong propensity for shallow landslide initiation on south-facing hillslopes in the study area and launched a research proposal. Miss Yanglin Guo completed the sampling collection and indoor tests.

**Competing interests**

All authors have declared that there were no conflicts of interests and competing interests.

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
