# Peer review of "Elucidating the role of soil hydraulic properties on aspect-dependent landslide initiation"

_EGUsphere, 2022_

## Author Comment (AC1)

Manuscript: https://doi.org/10.5194/egusphere-2022-798-RC2

**Elucidating the role of soil hydraulic properties on the aspect-dependent**

**landslide initiation**

**Detailed Response to the Reviewer Comments**

We sincerely thank the editor and reviewers for their constructive comments, helping to improve the quality of our manuscript. We are sorry for delaying answer the valuable comments because we are asked to stay at home for almost 2 months, due to the Coronavirus policy in Beijing and Beijing Forestry University. In such a condition, the team members read the comments and revised carefully after we are permitted to enter the office. With full consideration of the reviewers' suggestions, the manuscript has been carefully reshaped and we made point-by-point responses to address the comments of the reviewers. In the following, we answer all comments (set in black fonts) and give response (set in blue fonts). Quotes of the revision are set using a red font.

**Reviewer #1:**

1. The paper investigates the question whether land-sliding initiation at slopes is aspect-dependent. The obvious reason for potential differences - the radiation budget - is not discussed explicitly, but resulting processes - differences in plant root strength, the pore water pressure (or larger evapotranspiration at the south-exposed slope) are.

**Response:** Thanks for your suggestions.

The main reason for the difference of slope aspect in this article is the differential weathering of bedrock caused by solar radiation, which makes the rock and soil mass show differences in the hydrological properties of the slope. Because the vegetation on the sunny and shady slopes in this area is larch, which belongs to shallow root plants. The length of root system is about 0.4m, and it does not reach the sliding layer. Therefore, the strength of root system is not a factor that causes the slope direction difference of landslide.

In the introduction part, we add a summary of the impact of solar radiation and vegetation roots on landslides to guide the main line of the article.

In some semi-arid environments of the Northern hemisphere, aspect-dependent landslide initiation during some extreme rainstorm events would provide valuable insight into the relative importance of different factors in developing accurate landslide susceptibility models (Ebel, 2015; Rengers et al., 2016; Li et al., 2021; Deng et al., 2022). These events provide thoughtful understanding about the amount of direct sunlight translate into differences in vegetation community, bedrock weathering, and soil development process (Fu, 1983; Wang, 2008; Bierman and Montgomery, 2014). These typical earth surface process indirectly affect hillslope hydrology and landscape dissection on hillslope scale. Importantly, rainfall-induced shallow landslides are one of the geomorphic agents on hillslope scale and governed by multiple factors, including hydrology, hillslope materials, bedrock underneath and the vegetation (Birkeland, 1999; Geroy et al., 2011; Lu and Godt, 2013). Currently, the observed aspect-dependent landslide initiation mainly attributes to the mechanical effect of plant roots, because the differences of vegetation on the south- and north-facing slope are easier to examine and more obvious than other factors (Li et al., 2021; Timilsina et al., 2021; Dai et al., 2022; Deng et al., 2022). However, it is no denying that vegetation succession is far slowly than the soil development and bedrock weathering (Watakabe and Matsushi, 2019), and their roots in most cases is not deep enough to penetrate into bedrock (Schwinning, 2010). Hypothesizing in a localized area with same ecosystem or plant species, aspect-dependent landslide initiation cannot attribute to plant roots, while may result greatly from the differences in properties of hillslope materials due to long-term differential weathering.

The aspect-dependent landslides in Frontal Colorado, USA and the Loess Plateau, China have attracted interesting focus that vegetation generates a considerable influence on the landslide distribution. In fact, the overwhelming propensity for shallow landslide initiation on south-facing hillslope in the two regions closely relates to the present-day tree density, regardless of hillslope aspect (Ebel, 2015; Rengers et al., 2016; Deng et al., 2022). In the Colorado Frontal Range, field observations proved that southfacing slopes lack thick tree cover and have an abundance of rock outcrops compared to north-facing slopes, and the soil layer would be thinner on south-facing slopes (Coe et al., 2014; Ebel et al., 2015). The apparent cohesion supplied by roots was responsible for the observed connection between landslide distribution and slope aspect (McGuire et al., 2016). In the Loess Plateau China, vegetation recovery is the major ecological measure to mitigate the sediment loss (Fu et al., 2009). Promoted soil strength and hydraulic conductivity due to strong root network may enhance the topographic initiation condition (Montgomery and Dietrich, 1994; Wang et al., 2020). Another possibility is that the north- and west-ward moving storm produced more intense rainfall on south- and east facing slope. Such assumption may be invalid if aspect-dependent landslide distribution exists in a localized catchment with given vegetation communities. In fact, the above-mentioned study highlights the effect of mechanical function of plants on landslide. If the aspect-dependent landslide exists in a localized area that are covered by same plant species and high vegetation coverage, the observed aspect-dependent landslide initiation cannot attribute to mechanical effect from plant roots.

To elucidate the observed relationship among vegetation, landslide and slope aspect, the effect from physical properties and strength of hillslope materials cannot be ignored. In the Northern part of Loess Plateau China, as well as in many other semi-arid environments, different types and densities of vegetation and soils develop on north-facing versus south-facing convergent slopes, because systematic differences in the amount of direct sunlight translate into differences in the physical and chemical weathering. North-facing convergent slopes have lower evaporation rates, retain snow cover longer in spring, and tend to hold soil moisture longer into the summer growing season. Such differences may result in local ecosystem communities in presence of trees or shrubs over grasses. South-facing slopes experience heavier and more frequent hydration, thermal expansion or freeze-thaw cycle by the day warming and night cooling, and tend to favor stronger weathering throughout the year. Such differences could result in local differences in grain component, soil strength and soil profile, which indirectly affect the landslide scale by mechanics of excessive pore water pressure

dissipation and sliding surface liquefaction (Terzaghi, 1950; Sassa, 1984), and the hillslope hydrology behavior (Godt et al., 2009; Lee and Kim, 2019). Therefore, the physical properties of hillslope materials may attribute to the observed aspect-dependent landslide initiation.

2. The reviewer wants to know what exactly is your definition of north and south facing - which angle ranges (relative to North) is "allowed"? From the GoogleEarth images, there might very well also be west- and east-facing slopes. Were they excluded from the analysis?

**Response:** Thanks for your valuable comments.

In Section 4.1 of the result analysis, we added the definitions of south facing and north facing slopes, excluding the landslides on the east and west slopes.

In the study area, the south-facing slope is between 157.5 ° and 247.5 °, the north-facing slope ranges from 0 ° to 67.5 °, and 292.5 ° to 360 ° (0 ° is the due north). There were 71 shallow landslides on south-facing slope, while merely 20 landslides on north-facing slope.

3. The mechanisms leading to landslides are considered to some detail, including pore water dissipation, water storage and drainage, and stability fluctuations. The description of the latter is incomplete; there are equations shown for the "finite slope model" and the "infinite slope stability model", involving many parameters (such as angles) usually not easily to obtain in the field. Were these modelled by Hydrus-1D?  How reliable are these estimates?

**Response:** Thanks for your detailed comments.

For the parameter problems in the infinite and finite slope models, the slope and area of the landslide are extracted with high-precision topographic data using GIS software, the landslide depth is measured by field survey, the physical and mechanical property parameters are measured by GDS triaxial, and the hydrological parameters are measured by Trim combined with Hydras-1D. As for the reliability of the data of Hydrus-1D inversion, we made a supplementary explanation in Section 4.4.2 of the

conclusion, and proved the reliability of the parameters through the algorithm and optimization index used by the software.

The hydraulic properties, such as Soil Water Characteristic Curve (SWCC) and Hydraulic Conductivity Function (HCF), are critical to the analysis of water flow movement and mechanical behavior unsaturated soil material. In this study, the unsaturated hydraulic property measurement adopted Transient Release and Imbibition Method (TRIM). The intelligent advantage of TRIM method lies in that it combines physical and numerical experiments. In detail, it employs the simple and reliable measurement of transient water content by electronic balance to record the signature of transient unsaturated flow, and takes advantage of the robust inverse modeling capability to simulate the physical process. The apparatus can accommodate both undisturbed and remolded samples. The results of this study were obtained by using the Hydrus-1D code with the reverse modeling option, which implemented the Levenberg-Marquardt non-linear optimization algorithm, and minimized the error between the results in test and the simulation (Wayllace and Lu, 2012). Meanwhile, in order to ensure the uniqueness of the parameters, the aforementioned algorithm repeatedly run with different initial parameter estimates, until it always converges to the same or similar results. Compare the prediction results with the function curves of water flow and time obtained from the actual experiment, so that they can be basically combined to meet certain accuracy requirements. In this experiment, the R square of the regression between the optimized predicted value and the observed value is greater than 0.99. In addition, the model constraint effect of trim under two suction increment steps is better, and the parameters obtained by inversion calculation are more accurate (Lu and Godt, 2013). Table 2 shows the soil characteristic parameters obtained by Hydrus 1-D inversion.

4. Obviously (Fig. 11), the two Fs (eqs. 8 and 9) are dynamic quantities - which of the variables on the rhs are time-dependent? At least for the finite model, the time dependence seems to be marginal, but the difference in the mean values seems to be huge in comparison (Fig. 11 lower left panel). What is precisely the origin of

this discrepancy? The lower right panel shows that for the south-facing slope, the situation is rather similar (very stable values for Fs), whereas for the north-facing, Fs varies a lot. Why is that the case, i.e. what property or variable of eq. 9 is responsible for that? The reviewer disagrees with the statement that (thus) the infinite slope model would better support the observations, since the only rationale for that is by confirming the prejudice that south-facing slopes are more prone to landslides than north-facing ones. This is circular reasoning.

**Response:** Thanks for your detailed comments.

First of all, we add such a paragraph to the result 4.6 when we analyze the reasons for the difference in aspect using infinite and finite slope models. The purpose of finite slope model is to verify the problem from the mechanical point of view, while the infinite slope model is to verify the problem in the aspect of slope hydrological movement, because the mechanical properties, slope scale and bedrock permeability of north-south landslides are very different.

In this study, the infinite slope model and the finite slope model are used to characterize the sensitivity of landslide triggering, so as to determine the main mechanism of overwhelm landslide probability on south-facing slope. The infinite slope model studies the transient stress changes caused by water entering the soil, emphasizing the difference of soil permeability (Lu and Likos, 2006; Lu and Godt, 2013). The finite slope model focuses on the cohesion of the base surface and lateral periphery of the ground landslide source body, as well as the influence of the lateral additional cohesion provided by the vegetation root system for the landslide (Schmidt et al., 2001; Dai et al., 2022).

Secondly, in the two models, the time related parameter is saturation, which is calculated according to the actual monitored water content. The small correlation between the curve and time of the finite slope model may be due to the fact that the quantitative sliding area is considered in the calculation of the sliding force and resistance in the model, and the increment and fluctuation of the curve are reduced in the calculation; Secondly, the large difference between the north and south slopes in

the finite model mainly depends on the cohesion. The cohesion of the sliding layer soil on the sunny slope is twice that on the shady slope, so the stability of the sunny slope is higher. In the infinite model, the stability calculation of yin and yang slopes brings in the intake value and the parameters related to the aperture, such as α And n, these two parameters will affect the height and amplitude of the curve, mainly acting on the change of the suction stress of the slope during rainfall, which can characterize the movement of water through the pores after entering the soil mass, and may be the result of the joint action of macropores and matrix pores, which is the future research direction. This study is mainly to verify that the hydrological movement of the slope is the main reason affecting the difference of slope aspect by comparing the stability analysis results with the actual landslide density. In section 4.6 of the results, we added a paragraph describing the analysis results of the two models.

In all, the results of finite slope model reveal that the south-facing slope has relatively high stability, which mainly attribute to the fact that the effective cohesion of hillslope materials on the south-facing slope is stronger than that on the north-facing slope, even though the basal area of the landslide is more than twice. However, this result is inconsistent with the overwhelm landslide density on south-facing slope. Results of infinite slope model, considering the soil characteristic parameters of the soil moisture characteristic curve, reveal that the north-facing slope shows higher stability. In the analysis of finite and infinite models, the stability fluctuation amplitude of the south-facing hillslope is smaller than that of the north-facing hillslope, indicating that the water movement in the south-facing slope is less active than that in the north-facing slope. Therefore, in this study area, the change of soil stress is more sensitive to the slope stability than the change of root soil cohesion. It is verified that the change of soil permeability caused by differential weathering of bedrock could be responsible for the aspect-dependent landslide initiation in the study area.

5. However, the fundamental problem of the paper is that there is only a single site investigated, where landslides have occurred both at north-facing as well as south-facing slopes. The statistics of that particular location shows that more landslides

for south-facing slopes have been recorded: 71 versus 20 - this is probably the result of a field survey in the area, but the time span over which these happend should be mentioned as well, if known.   The whole area is densely tree covered, with a larch species dominating. In the GoogleEarth image, it seems that there are a lot of terraces surrounding the local peaks - is that due to management? If so, what was done there? This striking feature is not mentioned in the manuscript but could of course impact on landslide probability (either way).

**Response:** Thanks for your valuable comments.

In this study, all landslides were generated in July 2013. We purchased two high-precision remote sensing images in 2013 and 2010 to extract the landslides in the study area, and obtained the quantitative characteristics of landslides. As for the problem of terraces in the study area, we did not express it clearly when we discussed the study area before. Our study area is only in the Majiaba watershed in the north of Niangniangba, which is mainly mountainous. Terraces will only appear in the east and south of Niangniangba. The landslides we studied are on the slopes of the selected watershed, which can exclude the impact of man-made changes in land use types on landslides in the study area. Finally, we added the description of Majiaba watershed in the overview of the study area, and explained the reason for choosing it as the research object.

The study area is in the mountain region of Majiaba village in the northeast of Niangniangba town, Tianshui City, Gansu Province, Central China. It is also close to the dividing crest of the Yellow River and Yangtze River, and in the eastern part of the Loess Plateau. Majority of the hillslope are underlain by slate; the stratigraphic units of granite, sandstone, and mudstone account for a relatively smaller area. This area in semi-humid climate region and has four distinctive seasons. The annual precipitation is approximately 491.6 mm and mostly falls during June and August. One branch fault of the Tianshui-Lanzhou fault system runs through the area and has no rupture records for the last few decades.

The shallow landslides in the study area and nearby surroundings were triggered by the prolonged antecedent precipitation during 1 to 24 July and the intensive rainstorm on 25 July 2013 (Yu et al., 2014; Guo et al., 2015). Previous studies found that majority of shallow landslides in the whole storm-spanned mountain area have gradient of 20–25°, locate on south-facing slopes and in areas with sparse vegetation (Li et al., 2021). Besides, the strong root network may promote the hydraulic conductivity of soil-root composite and the landslide initiation condition of upslope contributing area-slope gradient, according the landslide cases in the *Larix Kaemphferi* and *Pinus tabuleaformis* forest (Dai et al., 2022). In this work, the three small catchment areas in Majiaba watershed are underlain by granite unit. The total area is 0.88 km$^2$ with vegetation coverage rate of over 90% (Fig. 1). The relative relief is about 200 m and the mean hillslope gradient is 37°. The reasons why choose the three catchments lie in that the main plant species on the south- and north-facing slope is *Larix Kaemphferi*, which commonly have highly-developed lateral roots with depth < 0.4 m. However, landslides in the three catchments still exhibit overwhelm propensity on south-facing slope over north-facing slope. Such a finding differentiates from the results in Frontal Colorado, USA, and Central Loess Plateau where landslides commonly occur in sparsely vegetated area. Furthermore, the works of Li et al. (2021) merely addressed the relationship between landslide probability and vegetation coverage in a regional scale, while neglected the importance of the properties of hillslope materials in a localized scale. Therefore, we hypothesize that such observation in the study area may not result from the mechanical effect of plant roots, but from distinctive physical properties and strength of hillslope materials due to differential weathering.

6. However, how could an investigation at one particular area say something conclusive about 3aspect-dependent landslide probabilities in general?

**Response:** Thanks for your valuable comments.

For the Niangniangba area, previous studies on the macro regional scale found that the landslide in the Niangniangba area has different slope directions, which is

characterized by a large number and area of landslides on the south slope and a small number and area of landslides on the north slope. In order to analyze the triggering mechanism of landslide at the micro scale, we conducted a detailed landslide survey in the mountainous areas in the north of Niangniangba, such as Majiaba, Shangyao, Beiyugou and other watersheds in August 2021, and selected representative sites for sensor monitoring and sampling analysis. This study is mainly aimed at specific watersheds. From the perspective of mechanics and hydrology, it explains the mechanism of landslide generation, provides some research methods and ideas, and reveals that the hydraulic characteristics of rock and soil mass are also a factor that cannot be underestimated compared with the vegetation factors.

7. In that regard, the paper seems to be way too ambitious. To do justice to the paper, systematic differences between north- and south-facing slopes are investigated to some detail. The slopes are rather steep but not different between S- and N-slopes (Fig. 3 left panel); grain diameter distributions are rather similar (Fig. 5); the physical properties reveal some differences, in particular for the saturated hydraulic conductivity, which of course can imply different water routing during and after rainfall events. On the other hand, in the unsaturated domain, it is not obvious that there are any differences in the pF curves (Fig. 8); they look strikingly similar for the two slope aspects. The shear tests (Fig. 6), on the other hand, seem to indicate that the two slope types have different pore water pressure behaviour (NB the reviewer wonders what the legend of that figure ("Time (10-sec)") would mean? Do you intend to say that the time axis is in logarithmic units (to base 10)? It doesn't seem to make sense).

**Response:** Thanks for your detailed comments.

In terms of gradient, the average gradient of landslide initiation on the south and north slopes does not differ much, but considering the catchment area, the critical condition for spatial initiation on the south slope is lower than that on the north slope. However, the shear strength and permeability of soil are the main influencing factors. In terms of particle size distribution, the south slope shows advantages in fine particles.

The soil water characteristic curve in Figure 8 shows that the soil on the south slope has more hysteresis effect than that on the north slope. In the pore water pressure characteristics, "10-sec" is the time interval for the software to automatically read data, which has no special meaning.

We performed CU tests to obtain the effective cohesion, effective internal friction angle, and the pore pressure water dissipation curves. The soil sampling, with diameter 50 mm and height of 100 mm, were firstly saturated in a vacuum pump, then consolidated in the chamber of GDS apparatus by 50, 100, 150, and 200 kPa confining pressure and 10 kPa backpressure. During each test, the shearing rate set as 0.1 mm/min, the device automatically records one data every 10s.

8. However, cause and effect are totally unclear here: are these differences induced by the different aspects, or just geological properties of the area, or random variation due to small sample sizes?.

**Response:** Thanks for your valuable comments.

The main reason for the difference of slope direction in the study area is the differential weathering of bedrock caused by solar radiation, which makes the rock and soil mass show differences in hydraulic properties. Then, the problem is explained from the aspects of shear strength, pore water pressure, unsaturated permeability and stability. In view of the randomness of the samples, we added a paragraph of discussion in Section 3.2 Field Sampling Survey Method, which shows that we have done a certain amount of actual investigation work in the field, and obtained that the soil mass in this area does have obvious weathering differences. At the same time, we added a paragraph of discussion in the discussion link, which shows that our samples are representative without randomness, Secondly, differential weathering of bedrock is also the most obvious geological feature in this area.

To investigate the hillslope hydrology on the south- and north-facing slope, the Frequency Domain Reflectometry (FDR) soil moisture sensors were used in this work to record volumetric water content. To avoid the randomness of data caused by natural factors such as terrain and vegetation, a total of 16 shallow landslides were investigated,

to excavate soil profiles and take undisturbed soil sampling. Then, the sensors were implemented at depths of 30cm, 70cm, and 110cm on south- and north-facing slope, to monitor the volumetric water content during the rainy season 2021.

Additionally, this work mainly highlights the role of hydraulic properties on the landslide occurrence. Though the south- and north-facing slope are merely underlain by granite, the physical properties of hillslope materials, such as the excessive pore water pressure, strength of sliding mass, soil water storage and leakage, differentiates a lot. Such a finding cannot be random because the study area is selected on condition that it is far from the northern and eastern area where local soils are mainly from Loess deposits, and the study area of Li et al (2021) and Dai (2022), where the bedrock underneath differs greatly.

9. A technical problem is that the language quality has to be improved. There are a non-negligible number of grammar errors and incomplete sentences which inhibits comprehensibility at times. Before resubmission, this issue should be carefully addressed.

**Response:** Thanks for your suggestions.

We try our best to improve the written English during the first revision. Also, we want find help from someone who can polish our manuscript.

10. Summarizing, the observational investigation for the selected sites is profound, and the processes and phenomena considered are numerous. However, the presentation is incomplete and in part difficult to follow, and most importantly, the conclusions drawn from this small field study seem to be too far-fetched. The paper deserves a major revision.

**Response:** Thanks for your suggestions.

Thanks for your admission on our works. We did make great works to explain the aspect-dependent landslide from a new perspective of hydraulic conductivity, other than from plant roots.

---

## Author Comment (AC2)

Manuscript: https://doi.org/10.5194/egusphere-2022-798-RC2

**Elucidating the role of soil hydraulic properties on the aspect-dependent landslide initiation**

**Detailed Response to the Reviewer Comments**

Thank you, Tammo Steenhuis.

Thanks for your constructive comments, helping to improve the quality of our manuscript. We are sorry for delaying answer the valuable comments because we are asked to stay at home for almost 2 months, due to the Coronavirus policy in Beijing and Beijing Forestry University. In such a condition, the team members read the comments and revised carefully after we are permitted to enter the office. With full consideration of the reviewers' suggestions, the manuscript has been carefully reshaped and we made point-by-point responses to address the comments of the reviewers. In the following, we answer all comments (set in black fonts) and give response (set in blue fonts). Quotes of the revision are set using a red font.

**Reviewer #2:**

The authors present a complicated explanation for the greater number of bank failures on the south slope than on the north slope.

**Response:** Thanks for the appreciative comments.

I am not sure if the analysis is correct since hillslope stability is not my field. However, I know that when the soil becomes saturated, the hillslope could fail, given that roots do not keep it in place. Based on this simple principle, we can explain, based on the data given in this manuscript, the difference between the north and south-facing slopes in simple terms as follows:

1. The conductivity of the subsoil is greater on the north-facing slope than on the south-facing slope. Thus, the north-facing slope drained faster than the south-facing slope, and as shown in Figure 9, the soil on the north-facing slope does not saturate. In contrast, on the south-facing slope, the rainfall rate at some point is greater than the water that can be carried off laterally, and the soil saturates, as shown in figure

9. The saturation causes the soil strength to decrease, and failure occurs. Hence more failures on the south slope than on the north-facing slope.

**Response:** Thanks for your detailed comments.

The drainage ability of hillslope materials is merely one of the factors. In the study area, the aspect-dependent landslide initiation may result mainly from the hydraulic conductivity, which indirectly attribute to the difference weathering. However, shallow landslides result from multiple factors, including the strength of hillslope materials, hydraulic conductivity, slope profile, and topographic factors. As the hydraulic conductivity plays a more important role on the landslide distribution, we mainly examined its role on the landslide initiation. However, the stability analysis results of the two models combines multiple factors, including the suction stress, cohesion, friction, topographic slope, failure depth. Importantly, the excessive pore water pressure dissipation strongly proves the drainage ability of hillslope materials. Therefore, excessive pore water pressure, together with the stability analysis greatly improve the understanding and elucidation of the reason about aspect-dependent landslide initiation.

2. Figure 9 is hardly discussed in the manuscript. It is likely the most significant finding as it shows that the oil becomes saturated on the south slope while not on the north slope.

**Response:** Thanks for your valuable comments.

First, thanks for your important reminds here and we may neglect your finding because we mainly focus on the water storage and leakage process based on the observed soil moisture. Secondly, figure 9 merely shows the volumetric water content during the observation stage, and mainly supports the results of figure. 10. In the revised manuscript, we added new contents according to your suggestions:

In comparison, it is likely the most significant finding as it shows that the soil becomes nearly saturated on the south slope while not on the north slope. This implies that the soil water on the south-facing slope is difficult to drain because of more fine

grains and the slow pore water pressure dissipation. Besides, the stable soil moisture of layers No. 2 and 3 for both slopes may attribute to long dry seasons in the study area, and the daily rainfall amount > 30 mm on July 9 and 23 resulted in soil moisture increase for all slope layers.

3. On line 377, the authors write that "the saturated hydraulic conductivities by variable-head permeameter and TRIM methods coincide with each other, which together prove that the soil mass on north-facing slope has a relatively larger water infiltration. The amount of water infiltrated on a slope depends on the amount of rainfall and not the conductivity as long as it is greater than the rainfall rate. Moreover, laboratory-derived conductivity is a poor predictor for field hydraulic conductivity in the topsoil where plant roots and animal life provide vertical preferential flow paths.

**Response:** Thanks for your detailed comments.

Your comments here highlighted the importance of in-situ measurements on the hydraulic conductivity. We also want to carry out the field hydraulic conductivity test, which would be more reliable than the laboratory-derived conductivity. However, it is very dangerous to carry out field tests because there were no human beings there after the extreme events in 2013. However, the Test of TRIM method were carried out in the laboratory. Besides, it is time-consuming to carry out in-situ measurements because the tests of preferential flow path must consider multiple sites. Therefore, we choose to monitoring the soil moisture to check the soil water storage and leakage. Additionally, you can see from figure 9 that the preferential flows are in form of sequential flow, not the nonsequential flow. In future, if possible, we may continue to examine the preferential flow path and analysis the effect on the landslide initiation.

4. As I indicated before, I leave it up to the experts if the hillslope analysis is correct or not. It seems too complicated for the little information that is available on this site. The fact that the soil strength decreases greatly at the time the soil becomes saturated is important and is not well addressed. In addition, the fact that soil saturates should be stressed in the manuscript that claims to be a hydrologic analysis.

**Response:** Thanks for your valuable comments.

There is no doubt that the hillslope stability analysis is correct in this work. On basis of suction stress definition in the Soil mechanics for unsaturated soils, the soil strength is from two parts: the first part derives from the particle connection, and the second part comes from the capillary force depending on the soil moisture. If the soil becomes nearly saturated, the soil strength will reduce greatly because the matrix suction disappears. The reason why we choose two stability models to analyze the hillslope stability fluctuation lie in that the role of hydrologic properties on the aspect-dependent landslide initiation is more important than other factors. Therefore, some hydraulic properties, such as the hillslope material properties, unsaturated conductivity, excessive pore water pressure, soil water storage and leakage, must the clarified in advance to support the stability analysis.

---

## Referee Report (RR1)

**Review - Elucidating the role of soil hydraulic properties on the aspect-dependent landslide initiation, by Yanglin Guo and Chao Ma (Special Issue: Experiments in Hydrology and Hydraulics)**

Preliminary remark: this is a review of the revised version of the manuscript. I was not involved with the reviews of the original version.

This paper analyses soil properties within an area of shallow landslides in China using field evidence and lab measurements. It concludes that differences in landslide occurrence observed on north-facing vs. south-facing slopes are explained by differences in soil hydraulic properties.

The manuscript is mostly well written and documented; the revisions clearly have improved the paper. Since the editor invited a revised version I take it that the manuscript is within the particular scope of this Special Issue. However, I would like to draw the attention to some major and minor points of the study that need clarification. These are detailed in the general and specific comments below. Finally, some technical corrections are listed.

**General comments**

*Interpretation of lab results, seepage model and stability analysis*

The authors present different analyses to underpin their reasoning how the soil properties favor landslides on south-facing slopes (S-slopes) compared to north-facing slopes (N-slopes). They use soil physical properties determined in the lab, field observations of soil moisture, modelled water storage and drainage, and stability analysis. The results for the hydraulical properties appear to support the conclusions, in particular Table 1 and Fig. 6. The uncertainty of these estimates, however, is not reported or discussed. Given the rather small sample size, which admittedly is also attributed to the efforts of the extensive testing as done here, the uncertainty and its implications should be discussed.

For the soil moisture observations (Fig. 9), I think it is arguable whether the differences are significant and representative for the N-slopes and S-slopes. Field monitoring of soil moisture as done here is also very much influenced by the local conditions and particular installation, and interpretation of absolute values need to consider sensor calibration. The maximum value is observed for the sensor in layer 1, S-slope (Fig. 9), whereas all three layers reach a (little lower) maximum at the N-slope. This could also hint at a higher susceptibility for deeper infiltration, and thus higher pore pressure that triggers land sliding, at N-slopes.

The seepage and stability models (Figs. 10 and 11) appear to contradict the notion that S-slopes are more prone to failure, as the N-slope both reach higher soil water storage (Fig. 10), and lower safety factors (Fig. 11). The authors admit that in line 411 ("the south-facing slope has a relatively high stability"), but contradict that in lines 414-416 ("Considering the soil parameters of the soil moisture curve, the results of the infinite slope model have shown that the north-facing slope showed a higher level of stability"). Please explain this apparent contradiction better, and explain in more detail how the SMC is supposed to turn the results of the stability analysis upside down.

One aspect that I found missing from the discussion are the different depths of the slip layer at N-slopes and S-slopes (lines 211-213). At S-slopes, different material are reported above and below the slip layer at ~ 0.85 m, while the material appears to change more gradually over the
slip layer at 1.05 m at N-slopes (Fig. 4). How do you think the layering influences slope
stability, or slope hydrology? In turn, it would also be interesting to discuss how the interplay
of rainfall, topography, and hydraulic and mechanical soil properties could be determining the
depth of the slip layer.

*The role of vegetation*

The authors start with the hypothesis that effects of plant roots, which was found for aspect-
dependent landslide initiation in other studies, are not relevant in their case. They report that
the roots of the main plant species, *Larix kaempferi*, do not extend to depths greater than 0.4
m (lines 112-114), and are thus above the depths of failure of the observed landslides (line
470). They use this as a reason to investigate other possible causes for the observed
differences in landslide occurrence. Unfortunately, the authors do not provide a source for the
estimated root depths, and only provide very little additional information about the vegetation,
which makes it hard to judge their argumentation. Further information would be needed to
support the claim that vegetation cannot be an important factor.

For example, is the distribution of rooting depths different on N- and S-slopes? If north and
south slopes have different rainfall, weather, and soil conditions, this could also affect plants
and their root characteristics. Plants are individuals, so even when they are from the same
species, their root systems might be influenced by age, microclimate, and soil conditions as
well.

How is the distribution of plant heights? I am not a botanist, but some quick info on *Larix*
*kaempferi* seems to suggest that the tree can grow rather tall (up to 30 m), and the minimum
rooting depth is around 0.5 m[1]. I would expect that taller trees tend to develop a root system
towards greater depths. Is the root system similar to the European Larch, which has both a
shallow (for nutrient uptake) and a deep-reaching central part (for tree stability)?

Were root depths observed in the landslide scars? The photos in Fig. 4 show a number of
roots, at these pits at least. Unfortunately, the depth cannot be read clearly.

The authors write in section 2 that the landslides in the area mainly occured on south-facing
slopes where vegetation was "sparse" (line 107). Ddid landslides occur in clear terrain, or
were trees affected as well? How different are the stand densities on N- and S-slope (cf. lines
61/62: "different types and densities of vegetation and soils develop on north-facing versus
south-facing convergent slopes")? Are there other relevant plant species on either N- or S-
slopes, which could contribute to soil strength by their (deeper) root systems?

* * *
[1] https://plants.sc.egov.usda.gov/home/plantProfile?symbol=LAKA2; 2023-02-11

**Specific comments**

Lines 99-100: Are the soils made of Loess, or is it just situated in the larger area of the Loess Plateau? What are the soil types on the N and S slopes?

Lines 160-162: Eq (2) - I fail to find the part where this is used or discussed in the paper.

Lines 189-192: How was $c_l$ parameterized? Which value were chosen for root cohesion? $S_{sr}$ and $\tau$ are not defined.

Lines 201-202: Why is the definition of south-facing slopes not symmetrical around 180°, as is the definition for north-facing slopes around 0°? How does this definition affect the results?

Lines 218-220: The lines in Fig 3b, which is where you base this statement on, are questionable and should be checked (also see comment on Fig 3). The difference in upslope contributing area is not easily visible in the data points in the figure, and the regression lines seem to be far away from the data points. Is the statement thus actually supported by the data? And, you are looking at the upslope contributing area above the head scar. The landslides on the S slopes have a longer stretch, and the initiation does not necessarily have to be at the uppermost point; more likely, it will start further downslope. Is the contributing area still smaller for the south-facing slopes, if you determine it from the lower end of the landslides?

Line 239, Fig 3b: The regression lines neither fit the mean values nor the individual data points?

Line 250, Fig 4: The photos show a number of roots. Unfortunately, the depth cannot be read. Please indicate a scale. Weights/Porosity diagrams: The (non-linear!) interpolation of the point measurements is misleading. It is not known if there is gradual or abrupt change in these values over the profile. The porosity diagram does not match the numbers in Table 1.

Line 363, Table 2: The difference between $K_s^d$ and $K_s^w$ is strikingly high. What is the uncertainty of the estimate, and is that not the opposite from what would be expected? In the paper of Wayllace and Lu (2012; reference cited in manuscript), the reported Ks of all samples were lower in the wetting, not in the drying phase. Also, it should be discussed how the values compare to $K_s$ in Table 1.

Lines 369-370: In comparison with the porosities in Table 1, soil moisture also almost reaches saturation on N-slope in all layers. It could also slightly exceed porosity in layer 1 on S-slope, but the other layers remain below saturation in Fig. 9.

Line 418-419: "change in soil stress was more sensitive to slope stability than the change in root soil cohesion": A bit unclear, which results for soil stress you refer to, the stability analysis? And change in root soil cohesion was not investigated or discussed, just excluded a priori.

Lines 439-444: This discussion of the higher cohesion observed at S-slopes is a bit confusing, because you first cite literature that would support greater depth of the slip layer and smaller sizes, but the opposite was observed. I think you want to argue why cohesion is not the crucial parameter here, but this should be made clearer.

Line 441: "some statistical results": Please specify.

Lines 451-453: This appears to be the opposite of what Fig. 11 shows: Failure potential
reaches higher peak and is more fluctuating at N-slopes.

Line 476: "Rich in clay content": Clay content appear to be below 5 % in all samples (Fig. 5).
I am not sure if this already considered rich in clay. Is the silt content significant in this
context?

**Technical comments**

Lines 13-15: "Remote sensing information … shallower depth" – Was landslide depth
assessed with remote sensing or field observations?

Lines 108-109: Check sentence "The strong root network may promote […] the landslide
initiation condition of the upslope contributing area–slope gradient,"

Line 189, Eq (8): g and z are not italicized in the numerator

Lines 189 & 195: $F_s$ is used in both equations, which might be misleading

Lines 236-237: Different definitions of the whiskers exist, please provide complete
information.

Line 250, Fig 4: "Gain" should be "Grain"

Lines 283-284: "The results … were taken here" – Check sentence

Line 292, Fig. 6: The units of the X-axis are unclear. Does the graph start at $10^0 = 10$ s, or at
$10^0 = 1$ s? Please give unambiguous units (seconds); scale the numbers if needed.

Line 309: "south" would rather be "north"? At least the higher permeability and lower pore
pressure were observed at the N-slopes.

Line 327, Fig. 7: The units of the X-axis are unclear. Does the graph start at $10^0 = 10$ min, or
at $10^0 = 1$ min? Please give unambiguous units (seconds); scale the numbers if needed.

Line 335: Check sentence structure, and give a reference for the TRIM method.

Line 470: "These observations cannot be attributed to plant roots" Unclear, which
observations "these" are. Check this, and the previous sentences.

---

## Author Response (AR2)

**Comments from reviewer 1 and the point-to-point replies**

**Interpretation of lab results, seepage model and stability analysis**

Comment: The authors present different analyses to underpin their reasoning how the soil properties favor landslides on south-facing slopes (S-slopes) compared to north-facing slopes (N-slopes). They use soil physical properties determined in the lab, field observations of soil moisture, modelled water storage and drainage, and stability analysis. The results for the hydraulical properties appear to support the conclusions, in particular Table 1 and Fig. 6. The uncertainty of these estimates, however, is not reported or discussed. Given the rather small sample size, which admittedly is also attributed to the efforts of the extensive testing as done here, the uncertainty and its implications should be discussed.

Reply: Good comments. Thanks sincerely.

We should discuss the uncertainty and its implications.

We added detailed contents to discuss the uncertainty and its implications in the discussion part "One of the possible limitations of this work lies in that the representativeness of the moisture observation and the uncertainty. Considering the multiple factors influencing landslides, the study area is selected with same bedrock underneath and similar plant species. Then, the moisture observation sites were selected on condition that similar soil profile, landscape with majority of landslides and the common topographical conditions. Therefore, this finding cannot be random because the study area has been selected on the condition that it is relatively far from the northern and eastern areas where local soils are predominantly loess deposits, and the study areas of Li et al. (2021) and Dai (2022), where the bedrock underneath differs substantially."

Comment: For the soil moisture observations (Fig. 9), I think it is arguable whether the differences are significant and representative for the N-slopes and S-slopes. Field monitoring of soil moisture as done here is also very much influenced by the local conditions and particular installation. The maximum value is observed for the sensor in layer 1, S-slope (Fig. 9), whereas all three layers reach a (little lower) maximum at the N-slope. This could also hint at a higher susceptibility for deeper infiltration, and thus higher pore pressure that triggers land sliding, at N-slopes.

Reply: Your comments here is a good doubt.

First, we checked lots of profiles and then decide the monitoring site during field investigations. If not do so, the soil moisture observation cannot be representative. The moisture observation was decided by following three conditions, including similar soil profile and soil particle component, high landslide density, and the *A-S* condition fitness. Similar soil profile and soil particle component has been done during field investigations. However, it is time-consuming if the soil particle analysis is completely made indoor (at least the soil samplings must be taken). In fact, the differences of soil profile is significant because the south-facing slope has more silt while the north-facing slope is sandy. Then, moisture observation site is selected on condition that the landscape with majority of landslides. Landscape with more landslides mean that the topographical condition and the soil mass are susceptible for landslides. Finally, the *A-S* condition fit the scope of topographical conditions of most landslides.

Comment: The seepage and stability models (Figs. 10 and 11) appear to contradict the notion that S-slopes are more prone to failure, as the N-slope both reach higher soil water storage (Fig. 10), and lower safety factors (Fig. 11). The authors admit that in line 411 ("the south-facing slope has a relatively high stability"), but contradict that in lines 414-416 ("Considering the soil parameters of the soil moisture curve, the results of the infinite slope model have shown that the north-facing slope showed a higher level of stability"). Please explain this apparent contradiction

better, and explain in more detail how the SMC is supposed to turn the results of the stability analysis upside down.

Reply: Great. Thanks sincerely for your comments here.

Your comments here did provide great help for us to improve the quality of this work.

We revised here into following form: "In finite slope model, the results have shown that the south-facing slope has a relatively high stability. However, this result contradicts to the high landslide density on the south-facing slope in the study area. In fact, the finite slope model does not consider suction stress, and the effective cohesion of hillslope materials mainly affects the stability result. In contrast, the results of the infinite slope model asserts that the state of the stress of the soil or regolith is modified by infiltration and changes in soil matrix suction. Furthermore, the fluctuation in fig. 11d also proves that the role of infiltration of water into shallow soils and the subsequent pore-water pressure response at depth is critical to the understanding the transient conditions that lead to shallow slope failure, because the stability fluctuation amplitude of the south-facing hillslope was smaller than that of the north-facing hillslope."

Comment: One aspect that I found missing from the discussion are the different depths of the slip layer at N-slopes and S-slopes (lines 211-213). At S-slopes, different material are reported above and below the slip layer at ~ 0.85 m, while the material appears to change more gradually over the slip layer at 1.05 m at N-slopes (Fig. 4). How do you think the layering influences slope stability, or slope hydrology? In turn, it would also be interesting to discuss how the interplay of rainfall, topography, and hydraulic and mechanical soil properties could be determining the depth of the slip layer.

Reply: Good comments here.

We added contents to discuss it in the "Discussion part".

To be honest, the slip layer of shallow landslide is commonly on the low-permeability layer or the boundary between soil and bedrock underneath. Such soil profile can lead to positive pore pressure to trigger slope failure. However, such recognition does not coincide with the study area as shown in fig. 4. As the depth of landslides is shallow, the depth of pit may be not deep enough to reach the bed rock underneath. However, the fine content of soil mass on the south-facing slope sufficiently proves the differential weathering of the rock and soil. In other words, The south-facing slope has a high degree of weathering, and the silt content of the soil between 40-80cm of the soil layer increases significantly. However, there is no obvious difference in the soil structure between 40-110m of the north-facing slope soil layer, and the bedrock will be reached from the bottom.

The newly revised fig. 4 is shown as follows:

[Figure]

**The role of vegetation**

Comment: The authors start with the hypothesis that effects of plant roots, which was found for aspect dependent landslide initiation in other studies, are not relevant in their case. They report that the roots of the main plant species, *Larix kaempferi*, do not extend to depths greater than 0.4 m (lines 112-114), and are thus above the depths of failure of the observed landslides (line 470). They use this as a reason to investigate other possible causes for the observed differences in landslide occurrence. Unfortunately, the authors do not provide a source for the estimated root depths, and only provide very little additional information about the vegetation, which makes it hard to judge their argumentation. Further information would be needed to support the claim that vegetation cannot be an important factor.

Reply: You are right here.

In fact, we carried out plenty of root depth check work before we installed moisture sensors. In view of the depth of vegetation roots, we can provide some soil profiles, landslide trailing edge and side wall, as well as actual photos of root distribution to support.

[Figure]

*Larix kaempferi* uprooted (< 40 cm)

[Figure]

Landslide trailing edge (south-facing slope). The root layer only accounts for about 1/3 of the profile.

[Figure]

Landslide sidewall (north-facing slope). Near the middle and lower part of the landslide mass, the bedrock has been exposed.

[Figure]

Soil profile of granite: (a) and (c) are South-facing slope; (b) and (d) are north-facing slope.

[Figure]

Soil profile of sandstone: (a) South-facing slope; (b) north-facing slope.

[Figure]

Soil profile of slate: (a) South-facing slope; (b) north-facing slope.

Comment: For example, is the distribution of rooting depths different on N- and S-slopes? If north and south slopes have different rainfall, weather, and soil conditions, this could also affect plants and their root characteristics. Plants are individuals, so even when they are from the same species, their root systems might be influenced by age, microclimate, and soil conditions as well.

Reply: Your comments here are right.

However, local forest workers told us the plants are planted at the same age.

It is true that different slope aspect and different microclimates will affect the growth of vegetation. We collected relevant data from the local forest research institute (www.xlsly.com). The afforestation time is between 2001 and 2002, which belongs to seedling afforestation. The landslide occurred in 2013, and the tree age is about 12 years. When we investigated the local landslide in 2021, the root depth was about 0.4-0.5m, However, it belongs to shallow root plants, and the root depth is unlikely to exceed 0.4 m 8 years ago. Therefore, no matter how the slope direction affects the growth rate of vegetation, it has little influence on the essential cause of triggering landslide.

Comment: How is the distribution of plant heights?? I am not a botanist, but some quick info on Larix kaempferi seems to suggest that the tree can grow rather tall (up to 30 m), and the minimum rooting depth is around 0.5 m. I would expect that taller trees tend to develop a root system towards greater depths. Is the root system similar to the European Larch, which has both a shallow (for nutrient uptake) and a deep-reaching central part (for tree stability)?

Reply: We have the plant height and age data.

The mean height of the plant is about 9.1 m (9.1m±2.2m), and the age of *Larix kaempferi* in the study area was about 20 years (until the year of 2021), the plant density differentiates a lot (ranges form 10 to 48 in an area of 100 m$^2$). In general, the plant density is higher at softer landscape while is lower at steep slope.

The root system is typically tap type (as shown in following figure) and has no deep-reaching central part.

[Figure]

Comment: Were root depths observed in the landslide scars? The photos in Fig. 4 show a number of roots, at these pits at least. Unfortunately, the depth cannot be read clearly.?

Reply: Yes, roots can be observed in the landslide scars while it not deep enough to the failure plane.

The root depth is about 0.4m, and the sliding depth can reach 0.8~1m. The bedrock has been exposed during our field investigations. The root system in Figure 4 belongs to the surface root system on the side wall of the soil profile, which has been modified and improved to further highlight the depth of the soil layer.

Comment: The authors write in section 2 that the landslides in the area mainly occurred on south-facing slopes where vegetation was "sparse" (line 107).Did landslides occur in clear terrain, or were trees affected as well? How different are the stand densities on N-and S-slope (cf. lines 61/62: "different types and densities of vegetation and soils develop on north-facing versus south-facing convergent slopes")? Are there other relevant plant species on either N- or S slopes, which could contribute to soil strength by their (deeper) root systems??

Reply: Good comments here.

Previous studies merely addressed the vegetation coverage on the landslide density (using the *GRVI* index in given pixel and corresponding landslide number), and found that landslide commonly occur in low-coverage area. It means that landslides also occur in high-coverage area while the landslide density is relatively smaller than the low-coverage area. It is difficult to get the plant density data because the slope is too steep in the study area while we measured the plant density at another places (about 10km to the study area)

The mean height of the plant is about 9.1 m (9.1m±2.2m), and the age of *Larix kaempferi* in the study area was about 20 years (until the year of 2021). The root system is typically tap type (as shown in following figure) and has no deep-reaching central part. Importantly, the plant roots have maximum depth of 50 cm according to our field investigations. Other plant species belongs to shrubs and grass and we didn't know their names.

Comment: Lines 13-15: "Remote sensing information … shallower depth" – Was landslide depth assessed with remote sensing or field observations?

Reply: Sorry to miss "field investigations" here.

We revised it into "Remote sensing interpretation using the high-resolution GeoEye-1 image, digitalized topography and field investigations showed that landslides on south-facing slopes have a higher probability, larger basal area, and shallower depth than those on a north-facing slope."

Comment: Lines 99-100: Are the soils made of Loess, or is it just situated in the larger area of the Loess Plateau? What are the soil types on the N and S slopes?

Reply: The soil here is very interesting.

According to our field investigation, the soil type for the right side of the Majiaba river is loess while the left side (the landslide area as shown in fig. 1) is not. Because the mountain region of the study area is on the transitional area of Loess Plateau and the Yanngte River, it is not completely situated in the larger area of Loess Plateau. We didn't classify the soil types on the $N$ and $S$ slopes in the manuscript because the particle component here is enough to exhibit the soil type. According to the USDA soil type classification and the percentage of silt, sand and clay content of the soil mass, the layers 1 and 2 on south-facing slope are silt loam, and the layer 1 is sandy loam. The soils on north-facing slope are sandy loam.

Comment: lines 108-109: Check sentence "The strong root network may promote […] the landslide initiation condition of the upslope contributing area–slope gradient,"

Reply: Ok. This sentence seems to be a little misunderstanding.

The meaning of this sentence is that plant roots can increase the strength and hydraulic conductivity of root-soil composite, thus the topographical threshold will increase correspondingly.

Then, we revised it into "Besides, some works found that plant roots may increase the topographical initiation threshold of landslides because of their positive effect on the strength and hydraulic conductivity of soil–root composite"

Comment: Lines 160-162: Eq (2) - I fail to f Bind the part where this is used or discussed in the paper.

Reply: Ok. We should clarify the purpose of equation (2).

The purpose of equation (2) is very simple. We want to compare the pore water pressure increase and dissipation ratio, to demonstrate that the importance of pore water pressure on the landslide scale.

In the revised manuscript, we rewrite this part to clarify its purpose, as shown follows:

"As a high excessive pore water pressure and slow dissipation ratio could cause widespread Coulomb failure within the shear zone, it will influence the landslide scale. To compare the rate of rise and dissipation of pore water pressure during the CU test, the ratio is expresses as

$$i = \frac{p_{t+\Delta t} - p_t}{\Delta t} \tag{2}$$

where $i$ is the increase or dissipation ratio of the excessive pore water pressure, and $p_t$ and $p_{t+\Delta t}$ are the pore water pressures measured during the time interval of $\Delta t$. A higher $i$ indicates that the pore-water within soil mass drainage rapidly and the pore-water pressure will dissipate in a short time. In other words, the $i$ is a proxy representing the hydraulic conductivity."

Comment: Line 189, Eq (8): g and z are not italicized in the numerator

Reply: OK.

We have revised the equation 8, including z in equation 9.

Comment: Lines 189 & 195: Fs is used in both equations, which might be misleading

Reply: Ok. We revised them using two terms.

The revised Fs is shown as follows:

$$F_s' = \frac{c_l A_l + c_b A_b + A_b(\rho_s - \rho_\omega S_e)\, gz\cos^2 \beta \tan \varphi'}{A_b \rho_s gz \sin \beta \cos \beta} \tag{8}$$

$$F_s'' = \frac{\tan \varphi'}{\tan \beta} + \frac{2c'}{\gamma z \sin 2\beta} - \frac{\sigma^s}{\gamma z}(\tan \beta + \cot \beta)\tan \varphi' \tag{9}$$

Comment: Lines 189-192: How was cl parameterized? Which value were chosen for root cohesion? $S_{sr}$ and $\tau$ are not defined.

Reply: Ok. We should clarify in the text.

$\tau$ is the shear force and $S_{sr}$ is the resistance. Considering too much terms in the text, we deleted the $\tau$ and $S_{sr}$ in eq. (8).

For the $c_l$ and the value were chosen for root cohesion, we used the cohesion of layer 1 and layer 2 for $c_l$, and we clarified in the revised manuscript "$c_l$ is the effective cohesion along the sidewall (kPa) and adopts the cohesion of layer 1 and layer 2, $c_b$ is the basal soil cohesion (kPa), and adopts the cohesion of layer 3".

Comment: Lines 201-202: Why is the definition of south-facing slopes not symmetrical around 180°, as is the definition for north-facing slopes around 0°? How does this definition affect the results?

Reply: No. The definition of south- and north-facing slope cannon use 180° and 0° in the study area.

The study area is in Northern Hemisphere and the sunlight illumination does not coincide with the aspect orientation. In summer season, the position of the direct sunlight is between the equator and the Tropic of Capricorn, not between the equator and the Tropic of Cancer. In fact, the location of the study area is located north to the Tropic of Cancer.

Therefore, we wrote a sentence to define the south- and north-facing slope "In the study area, the direct sunlight does not coincide with the aspect orientation because it is in the north the Tropic of Cancer. The south-facing slope is defined between 157.5 ° and 247.5 ° and the north-facing slope is between 0 ° to 67.5 °, and 292.5 ° to 360 ° (0 ° is the due north)."

Comment: Lines 218-220: The lines in Fig 3b, which is where you base this statement on, are questionable and should be checked (also see comment on Fig 3). The difference in upslope contributing area is not easily visible in the data points in the figure, and the regression lines seem to be far away from the data points. Is the statement thus actually supported by the data? And, you are looking at the upslope contributing area above the head scar. The landslides on the S slopes have a longer stretch, and the initiation does not necessarily have to be at the uppermost point; more likely, it will start further downslope. Is the contributing area still smaller for the south-facing slopes, if you determine it from the lower end of the landslides?

Reply: Ok

In the caption of Fig. 3, we provided detailed information about the lines in fig. 3b.

The provided information is "The definitions of the whiskers are shown in caption of fig. 2. The circles are averaged slopes with the radius size proportional to the number of landslides. The small cross represent all individual data values. The power-law regression is fitted with the dataset closet to the axis origin."

The low limit of upslope contribution area and slope data to be fitted by the regression line is used as the critical starting condition for analysis.

The erosion ditch results from the displaced mass travel along the unchanneled valleys, which is not the contributing area. The contributing area starts from the head-scar (upper boundary), not from the lower end of the landslides.

Comment: Line 239, Fig 3b: The regression lines neither fit the mean values nor the individual data points?
Reply: Individual data points.

The low limit of upslope contribution area and slope data to be fitted by the regression line is used as the critical starting condition for analysis. We added more information in the caption of fig. 3 "**Fig. 3.** Upslope contributing area and slope gradient condition: (a) Upslope contribution area and mean slope vs slope aspect; and (b) the upslope contributing area vs mean slope gradient above the landslide area. The definitions of the whiskers are shown in caption of fig. 2. The circles are averaged slopes with the radius size proportional to the number of landslides. The small cross represent all individual data values. The power-law regression is fitted with the dataset closet to the axis origin."

Comment: Lines236-237: Different definitions of the whiskers exist, please provide complete information
Reply: Ok

We provide the complete information of the definitions of the whiskers in the caption of fig. 2.

The provided complete information is "The edge line of box in the box chart shows the $75^{th}$ quantile (Q3), median and $25^{th}$ quantile (Q1) from top to bottom. The length of the box is referred to as the inter-quartile range (IQR). The crossed square inside the box is the average value. The whiskers extend to the maximum and minimum values except the mild outliers. The upper limit and lower limit of whiskers are Q3+1.5IQR and Q1-1.5IQR respectively. The circles are the outliers, and the cross symbol is the maximum and minimum values for all the data."

Comment: Line 250, Fig 4: The photos show a number of roots. Unfortunately, the depth cannot be read. Please indicate a scale. Weights/Porosity diagrams: The (non-linear!) interpolation of the point measurements is misleading. It is not known if there is gradual or abrupt change in these values over the profile. The porosity diagram does not match the numbers in Table 1.
Reply: Ok.

The roots are surficial and not penetrate the failure plane. We revised figure 4 (extending the depth line to the left figure) to ensure the depth can be read. Besides, we made a mistake that the porosity diagram doesn't match the numbers, and we revised them.

[Figure]

Comment: Lines 250: Fig 4: "Gain" should be "Grain"

Reply: Done, Sorry to make mistake here.

In the revision process, we checked such word errors throughout the manuscript.

Comment: Lines 283-284: "The results … were taken here" – Check sentence

Reply: Done

We want to say "compare the results of the pore water pressure during the consolidation process under 200 kPa effective confining pressure". Therefore, we revised it into "The results of the pore water pressure during the consolidation process under 200 kPa effective confining pressure were compared here".

Comment: Line 292, Fig. 6: The units of the X-axis are unclear. Does the graph start at 100= 10 s, or at 100= 1 s? Please give unambiguous units (seconds); scale the numbers if needed. Lines 160-162: Eq (2) - I fail to bind the part where this is used or discussed in the paper.

Reply: Done

We provided clear information of the fig. 6 and added detailed description in the caption.

The added content is "The values in the figure 6 are the average rates of rise and dissipation of pore-water pressure during consolidation calculated by Equation 2. The GDS software automatically reads a measured pore pressure data every 10 s, so the time starts from $10^0$= 10 s of the *x*-axis."

Besides, we mentioned that and the device automatically recorded data every 10 s (Line 154).

Comment: Line 309: "south" would rather be "north"? At least the higher permeability and lower pore pressure were observed at the N-slopes.

Reply: Done

We are sorry to make a mistake here. May be wrong description here.

We revised it into "Therefore, the high permeability of the soil mass on the north-facing slope may result in low peak pore water pressure."

Comment: Line 327, Fig. 7: The units of the X-axis are unclear. Does the graph start at 100= 10 min, or at 100= 1

min? Please give unambiguous units (seconds); scale the numbers if needed.

Reply: OK.

The measured duration during the test is too long. It's better to use minutes than seconds.

The software record data 10 minutes a time. So we use 10 minutes as unit in the axis.

We provided detailed information in the caption "The software automatically records the mass of water outflow 10 min each, so the $x$-axis starts from $10^0$"

Besides, we mentioned that the water outflow mass was measured on a 10 min basis in the part of "3.3 Water storage and drainage"

Comment: Line 335: Check sentence structure, and give a reference for the TRIM method.

Reply: OK.

We revised it into "The Soil Water Characteristic Curve (SWCC) and Hydraulic Conductivity Function (HCF) are critical for the analysis of water flow movement and mechanical behavior of unsaturated soil material. In this study, the Transient Release and Imbibition Method (TRIM) for unsaturated hydraulic property measurement (Lu and Godt, 2013)."

Comment: Line 363, Table 2: The difference between Ksd and Ksw is strikingly high. What is the uncertainty of the estimate, and is that not the opposite from what would be expected? In the paper of Wayllace and Lu (2012; reference cited in manuscript), the reported Ks of all samples were lower in the wetting, not in the drying phase. Also, it should be discussed how the values compare to Ks in Table 1.

Reply: Good comments here.

The uncertainty factor affecting the estimated value is the distribution and quantity of soil pores, especially the existence of soil macropores, which will significantly affect the soil permeability rate. For undisturbed soil, the soil texture is uneven. It is necessary to consider the particle composition of the soil, and the existence of cracks, number of roots. Although Trim test measures the permeability of soil matrix, which is mainly related to particle composition, the influence of other seepage channels on the permeability coefficient cannot be ignored, and the contribution of such influence on the permeability rate cannot be measured at present. However, the current inversion data results are consistent with the expected results. The saturated permeability rate of the north-facing slope is higher than that of the south-facing slope. In the paper of Waylace and Lu (2012; references cited in the manuscript), the permeability coefficient of all samples Ks in the moisture absorption stage is lower than that in the dehumidification stage, but he also only presents the experimental results of remolded sandy colluvial soil, remolded silty clay and original silty clay, and the sample size itself is not particularly large, and the homogeneity of remolded soil is certainly greater than that of the original soil, and in remolded sandy colluvial soil, in fact, the permeability data of moisture absorption and dehumidification are not very different.

In the revised version, we added some discussion in the "Discussion part": "However, the difference b between $K_s^d$ and $K_s^w$ is strikingly high and the $K_s^d$ is smaller. Although Trim test in this work measures the permeability of soil matrix, the influence of other seepage channels on the permeability coefficient cannot be ignored, and the contribution of such influence on the permeability rate cannot be measured at present."

Comment: Lines 369-370: In comparison with the porosities in Table 1, soil moisture also almost reaches saturation on N-slope in all layers. It could also slightly exceed porosity in layer 1 on S-slope, but the other layers remain below saturation in Fig. 9.

Reply: No. Saturation condition is temporal and disappears at two peak rainfall stage during the monitoring stages.

The saturation condition temporally occurs at two rain events with hourly rainfall close to 30mm/h. Such

high hourly rainfall may result in preferential flow, which mainly occur in macro-pores. However, the moisture fluctuation can prove that the water infiltration and drainage process on south-facing slope is lower than the north-facing slope.

Comment: Line 418-419: "change in soil stress was more sensitive to slope stability than the change in root soil cohesion": A bit unclear, which results for soil stress you refer to, the stability analysis? And change in root soil cohesion was not investigated or discussed, just excluded a priori. "
Reply: Yes. We make a mistake here.

Not soil stress, but suction stress. Make a wrong writing here.

We revised this sentence into "Therefore, in the study area, the change in soil suction stress was more sensitive to slope stability than the change in root soil cohesion."

Comment: Lines 439-444: This discussion of the higher cohesion observed at S-slopes is a bit confusing, because you first cite literature that would support greater depth of the slip layer and smaller sizes, but the opposite was observed. I think you want to argue why cohesion is not the crucial parameter here, but this should be made clearer. Line 441: "some statistical results": Please specify. "
Reply: Good comments here.

First, the larger landslide area and greater width of landslides on south-facing slope can be explained by the pore-water dissipation because the slow dissipation of excessive pore-water pressure because widespread liquefaction may cause extend the landslide scale.

Second, we try our best to discuss the reason why the slip layer of landslides on south-facing slope is thinner than that on north-facing slope, and found that it cannot be explained because the theoretical maximum or maximum slip layer for strong-cohesive slope should be larger than weak-cohesive slope at given slope (Iida, 1999; D'Odorico and Fagherazzi, 2003). One of the reasons may be that cohesive soil mass often hold tight together to displace downslope owing to the strength loss. The relatively weak-cohesive soil mass often loosens to displace downslope, with the slip layer may be close to the boundary between soil mass and bedrock underneath.

Therefore, we revised here to following forms and deleted previously confusing discussion: "This may be attributed to the fact that the slow dissipation of excessive pore-water pressure because widespread liquefaction may cause extend the landslide scale. For the thinner slip layer of landslides on south-facing slope, it may result from differential weathering, because the theoretical maximum or maximum slip layer for strong-cohesive slope should be larger than weak-cohesive slope at given slope (Iida, 1999; D'Odorico and Fagherazzi, 2003). One of the reasons may be that cohesive soil mass often hold tight together to displace downslope owing to the strength loss. The relatively weak-cohesive soil mass often loosens to displace downslope, with the slip layer may be close to the boundary between soil mass and bedrock underneath."

Comment: Lines 451-453: This appears to be the opposite of what Fig. 11 shows: Failure potential reaches higher peak and is more fluctuating at N-slopes.
Reply: No, No, No.

Higher failure potential means the stability always approaches to 1.0. If the stability is far over 1.0, it means less failure potential. The more fluctuation of stability at N-slope is reasonable because the water infiltration and drainage is faster than S-slope.

Comment: Line 470: "These observations cannot be attributed to plant roots" Unclear, which observations "these" are. Check this, and the previous sentences. "

Reply: Sorry to make a mistake here.

We revised it into "Such overwhelming landslide phenomenon cannot be attributed to plant roots and may result from the differential weathering of bedrock under the influence of hydrothermal conditions."

Comment: Line 476: "Rich in clay content": Clay content appear to be below 5 % in all samples (Fig. 5). I am not sure if this already considered rich in clay. Is the silt content significant in this context? "

Reply: You are right here, we adopt your comments here.

We revised it into "the soil masses on the south-facing slope have higher silt content than those on the north-facing slope."

Other revisions:

(1) Considering the labels of *x*-axis in figs. 9-11 relate to time and date respectively. We improved the unit of the *x*-axis of the three figures. Please see the revised manuscript. Besides, the unit of the *x*-axis in fig. 8 is wrong, and is dimensionless. We revised it.

(2) some references were inserted and some deleted

Inserted references:

Iida, T.: A stochastic hydro-geomorphological model for shallow landsliding due to rainstorm. Catena, 34(3-4), 293-313, https://doi.org/10.1016/S0341-8162(98)00093-9, 1999.

D' Odorico, P., Fagherazzi, S.: A probabilistic model of rainfall-triggered shallow landslides in hollows: A long-term analysis, Water Resour. Res., 39(9), 1262, https://doi.org/10.1029/2002WR001595, 2003.

Maier, F., van Meerveld, I., Greinwald, K., Gebauer, T., Lustenberger, F., Hartmann, A., Musso, A.: Effects of soil and vegetation development on surface hydrological properties of moraines in the Swiss Alps, Catena, 187, 104353, https://doi.org/10.1016/j.catena.2019.104353, 2020.

Lohse, K. A., Dietrich, W. E.: Contrasting effects of soil development on hydrological properties and flow paths, Water Resour. Res., 41, 1–17, https://doi.org/10.1029/ 2004WR003403, 2005.

Deleted references:

Larsen, I. J., Montgomery, D. R., Korup, O.: Landslide erosion controlled by hillslope material, Nat. Geosci., 3, 247-251, https://doi.org/10.1038/ngeo776, 2010.

Frattini, P., Crosta, G. B.: The role of material properties and landscape morphology on landslide size distributions, Earth Planet. Sci. Lett., 361, 310-319, https://doi.org/10.1016/j.epsl.2012.10.029, 2013.

Milledge, D. G., Bellugi, D., McKean, J. A., Densmore, A. L., Dietrich, W. E.: A multidimensional stability model for predicting shallow landslide size and shape across landscapes, J. Geophys. Res.: Earth Surf., 119, 2481-2504, https://doi.org/10.1002/2014JF003135, 2014.

Comment: This paper presents an analysis of the difference in aspect-dependence of landslide initiation. The objective is to show the importance of hydraulic conductivity on the differences in behavior between north and south-facing slope. To eliminate other explanatory factors, a detailed description of the soils, vegetation and pore pressure was conducted. Several techniques were used to ensure the validity of the conclusions. I think that the conclusions on the role of the different soil parameters show clearly that the stability analysis must take them all into account to be accurate. However, I totally agree with the remarks of the previous reviewers concerning the representativeness and generalization of the results. Some points like the spatial distribution of the measurements, the difference between stability models, the importance of saturation in the upper layers have been improved. In the discussion, some elements were added to make the link with the different observations and the conclusions on the role of the face orientation. Perhaps the variability in the measurement of the properties of 16 soil profiles would make it possible to be more convincing about the difference in structure between North and South facing slope. However, it is not yet clear to me how aspect could have changed the hydraulic conductivity of the soil and especially if this is generalizable to other locations where the climatic variations between north and south face are not identical to this area.

Reply: Thanks.

Thanks for your admission.

Enough soil profiles support strong evidences of the differences in structure while some easy or smart ways can also be used during field investigations. Before we installed the sensors, we checked the soil profiles by test pits. In fact, some scholars of geomorphology and geology with rich experience on the soil type know that sandy soil are commonly frictional and easily washed away by water, while the silty and clayey soils are slippy and addictive (if such soil addicts on hand, it is difficult to wash way by water). The scholars in this work mainly study the landslides and debris flows across the landscape of China (such as Tibet, Loess Plateau, southeast part, northeast terrain, mountain regions near Beijing) and have lots of experience. We adopted this easy method before we select sites to install sensors.

Secondly, when we found the soil differences in the study area, it is so interesting and we decided with carry out further work here. The preliminary work enables us the choose the proper sites to install sensors and take sampling works for indoor tests. The reason why we choose the study area lies in that it is not too large and is helpful for us to check the soil profiles on south- and north-facing slope. However, soil profile data is not enough to analyze the aspect-dependent landslide initiation. The landslide occurrence is a result of hillslope hydrology, suction stress, and the stability. And either of them cannot be neglected. As you said, if more soil profiles data is available, the difference in structure between North and South facing slope is to be more convincing.

Thirdly, i think the contents of this work justly present the differences in hydraulic conductivity of the soil, while cannot explain why how the aspect change the hydraulic conductivity. If we highlight the how the aspect changes the hydraulic conductivity, it will beyond the main contents of this work, because it needs the solar radiation data of both slopes, dating of soil development time and regolith time. However, the data presented in this work give some evidence, such as the high fine content on south-facing slope over north-facing slope sufficient proves the high degree of weathering.

Comment: In particular, as I am not a specialist, I wonder how the soil alteration is caused by solar radiation: consequence of the vegetation, spatial distribution of precipitation (only one rain gauge), wind. In addition, the time scale on which this influence is thought to play a role should be specified.

Reply: Yes.

However, it is a very challenging work to address the soil alteration by solar radiation

What you said here coincide with the suggestions and comments from other scholars. We also find help from some scholars about the issues of aspect-dependent landslide initiation. Such phenomenon can be explained by lots of ways, such as the hillslope hydrology and stability, dating methods examine the soil development, even preferential flow in the macropore distribution.

Therefore, we added some descriptions to clarity the time scale is thought to play a role "These differences result from differential weathering owing to the amount of direct sunlight. Other methods such as numerical or relative dating methods and preferential flow in the macropore distribution could provide new evidence for such observations."

As far as the role of soil alteration by solar radiation, it is an interesting aspect while tremendous work needs to prove it. According to our experiences in Loess Plateau China, solar radiation is a important factor controlling the vegetation and soil moisture. First, slope aspect is a very important topographic feature, which has a significant impact on local climate, especially solar radiation, temperature, evaporation, water, etc., and will have a profound impact on vegetation, soil, and hydrology. The vegetation coverage on the south-facing slope is generally less than that on the north-facing slope, which is mainly composed of shrubs and herbs, while the north-facing slope is a combination of trees, shrubs, and grasses. The vegetation will affect the interception of rainfall, regulate the surface runoff, and the development of the root system will also affect the infiltration channel and the anchoring effect on the soil. The south-facing slope has large rainfall, high rainfall intensity, high temperature, large diurnal range, and the rain and heat in the same season jointly promote the formation of soil cracks, accelerate the physical and chemical weathering of rocks and soil, become a high incidence area of water-rock chemical action, and promote the formation of clay minerals. Due to the better light and heat conditions, human activities are more frequent on the south-facing slope, which affects vegetation coverage, accelerates soil erosion, and gradually evolves into collapse, landslide, and debris flow disasters.

Comment: Could a stability analysis on the 2013 event that caused several landslides be done using rainfall data (even approximated) to validate the assumptions.

Reply: I want to do it, while we didn't have the moisture data in the 2013 event.

What you said here justly coincide with our previous idea. However, if we use the monitored data to simulate the soil moisture in the 2013 event, the soil suction data must be monitored. Therefore, we must quit such simulation work because of no monitored soil suction data. Also, we may monitor the soil suction and soil moisture together in future to obtain physical parameters, then to simulate the suction stress and moisture in 2013 event.

**Detailed comments and reply**

Comment: Fig 1: elevation legend refers to which map. Landslips at the north face are not very visible.

Reply: Done.

Elevation legend refers to the mountain spanning Niangnaingba and Majiaba.

The large graph in fig. 1 is the highest resolution obtained by UAV. It is the most visible graphs though we have the 0.5 m resolution GeoEye-1 image (0.5 m × 0.5 m). As you said, landslips at the north face are not very visible, so we draw the boundaries.

We revised the Caption of figure 1 as "Fig. 1. Location, topography, and simplified lithology of the study area. All maps are created by the authors. The graph of Majiaba was taken using an Unmanned Aerial Vehicle. The territorial domain of China and simplified lithology map are from China Geological survey. Elevation legend refers to the mountain spanning Niangnaingba and Majiaba."

Comment: L125: obtaining DEM (LIDAR ?), figure 1 from a UAV at which elevation ?

Reply: Lidar DEM.

Because of high-resolution of the DEM is protected and cannot be shown according to the requirements of local surveying and mapping agency. Besides, we have the UVA-derived DSM and the map of the study area. The mountain region (containing the study area) ranges from 1329 and 2300m.

We added the detailed information about the elevation in "2 Study area": "The elevation of the mountains in the mountain region of the study area ranges from 1329 m to 2300 m."

Comment: Fig3b: the linear regression seems far from data.

Reply: It's power-law form and represents lower limit line.

We provided detailed information of caption of fig. 3b.

The revised caption is revised into "The definitions of the whiskers are shown in caption of fig. 2. The circles are averaged slopes with the radius size proportional to the number of landslides. The small cross represent all individual data values. The power-law regression is fitted with the dataset closet to the axis origin."

Comment: L290: values can be only read on the figure 6

Reply: Done.

We added detailed information in the caption of figure.

The caption is revised into "The values in the figure 6 are the average rates of rise and dissipation of pore-water pressure during consolidation calculated by Equation 2. The unit of $x$-axis marks the time record interval of 10 seconds."

Comment: Figure 6: put in caption the label meaning.

Reply: Ok

The revised caption of fig. 6 is "Variation in pore-water pressure under effective confining pressure of 200 kPa by GDS triaxial shear tests. The values in the figure 6 are the average rates of rise and dissipation of pore-water pressure during consolidation calculated by Equation 2. The unit of x-axis marks the time record interval of 10 seconds."

Comment: L336: I am not sure numerical experiments is the right term, does it mean calibration?

Reply: Yes, you are right here.

We revised it into "The advantage of the TRIM method is that it combines physical experiments and calibration."

Comment: Fig 11: put in caption what the pink line means.

Reply: Done

The pink lines indicate the stability equals to 1.0.

We revised the caption into "Change in slope stability fluctuation: (a) rainfall records, (b) degree of saturation, (c) stability of finite slope model, and (d) stability of infinite slope model. The pink lines indicate the stability equals to 1.0."

Comment: L462: I wonder if laboratory studies in the literature can confirm this remark.

Reply: It's a challenging work.

This aspect-dependent landslide initiation could be explained by either hydraulic conductivity work or the stability analysis. The two important aspects are involved in this work. At least, in our knowledge scope or some

cases that I know, elucidation of aspect-dependent landslide initiation must find ways from hydraulic conductivity work or the stability analysis.

Currently, the works addressing the aspect-dependent landslide initiation worldwide merely refer to the landslide cases in 2013 rainstorm Frontal Colorado range USA, where the plant roots have been resumed as the main controlling factor. Though some studies used numerical simulation works to explain the aspect-dependent landslide initiation there, i found local plant roots is not deep enough to penetrate over the failure plane. Therefore, we highlighted the importance of hydrological properties on both slopes to explain the aspect-dependent landslide initiation.

Comment: L465:" other methods" instead of mechanism?

Reply: Ok.

Your comments are better than previous words "mechanism".

We adopt your suggestions here and revised it into "Other methods such as numerical or relative dating methods and preferential flow in the macropore distribution could provide new evidence for such observations."

Other revisions:

(1) Considering the labels of *x*-axis in figs. 9-11 relate to time and date respectively. We improved the unit of the *x*-axis of the three figures. Please see the revised manuscript. Besides, the unit of the *x*-axis in fig. 8 is wrong, and is dimensionless. We revised it.

(2) some references were inserted and some deleted

Inserted references:

Iida, T.: A stochastic hydro-geomorphological model for shallow landsliding due to rainstorm. Catena, 34(3-4), 293-313, https://doi.org/10.1016/S0341-8162(98)00093-9, 1999.

D' Odorico, P., Fagherazzi, S.: A probabilistic model of rainfall-triggered shallow landslides in hollows: A long-term analysis, Water Resour. Res., 39(9), 1262, https://doi.org/10.1029/2002WR001595, 2003.

Maier, F., van Meerveld, I., Greinwald, K., Gebauer, T., Lustenberger, F., Hartmann, A., Musso, A.: Effects of soil and vegetation development on surface hydrological properties of moraines in the Swiss Alps, Catena, 187, 104353, https://doi.org/10.1016/j.catena.2019.104353, 2020.

Lohse, K. A., Dietrich, W. E.: Contrasting effects of soil development on hydrological properties and flow paths, Water Resour. Res., 41, 1–17, https://doi.org/10.1029/ 2004WR003403, 2005.

Deleted references:

Larsen, I. J., Montgomery, D. R., Korup, O.: Landslide erosion controlled by hillslope material, Nat. Geosci., 3, 247-251, https://doi.org/10.1038/ngeo776, 2010.

Frattini, P., Crosta, G. B.: The role of material properties and landscape morphology on landslide size distributions, Earth Planet. Sci. Lett., 361, 310-319, https://doi.org/10.1016/j.epsl.2012.10.029, 2013.

Milledge, D. G., Bellugi, D., McKean, J. A., Densmore, A. L., Dietrich, W. E.: A multidimensional stability model for predicting shallow landslide size and shape across landscapes, J. Geophys. Res.: Earth Surf., 119, 2481-2504, https://doi.org/10.1002/2014JF003135, 2014.